# N6-methyladenosine regulated FGFR4 attenuates ferroptotic cell death in recalcitrant HER2-positive breast cancer

Yutian Zou [1,5], Shaoquan Zheng[1,5], Xinhua Xie[1,5], Feng Ye [1], Xiaoqian Hu[2], Zhi Tian[3], Shu-Mei Yan[1], Lu Yang [1], Yanan Kong [1], Yuhui Tang[1], Wenwen Tian[1], Jindong Xie[1], Xinpei Deng[1], Yan Zeng[1], Zhe-Sheng Chen [4✉], Hailin Tang [1✉] & Xiaoming Xie [1✉]

Intrinsic and acquired anti-HER2 resistance remains a major hurdle for treating HER2-positive breast cancer. Using genome-wide CRISPR/Cas9 screening in vitro and in vivo, we identify FGFR4 as an essential gene following anti-HER2 treatment. FGFR4 inhibition enhances susceptibility to anti-HER2 therapy in resistant breast cancer. Mechanistically, m6A-hypomethylation regulated FGFR4 phosphorylates GSK-3β and activates β-catenin/TCF4 signaling to drive anti-HER2 resistance. Notably, suppression of FGFR4 dramatically diminishes glutathione synthesis and $Fe^{2+}$ efflux efficiency via the β-catenin/TCF4-SLC7A11/FPN1 axis, resulting in excessive ROS production and labile iron pool accumulation. Ferroptosis, a unique iron-dependent form of oxidative cell death, is triggered after FGFR4 inhibition. Experiments involving patient-derived xenografts and organoids reveals a synergistic effect of anti-FGFR4 with anti-HER2 therapy in breast cancer with either intrinsic or acquired resistance. Together, these results pinpoint a mechanism of anti-HER2 resistance and provide a strategy for overcoming resistance via FGFR4 inhibition in recalcitrant HER2-positive breast cancer.

[1] Sun Yat-sen University Cancer Center; State Key Laboratory of Oncology in South China; Collaborative Innovation Center for Cancer Medicine, Guangzhou, China. [2] School of Biomedical Sciences, Faculty of Medicine, The University of Hong Kong, Hong Kong, China. [3] College of Pharmacy, University of South Florida, Tampa, FL, USA. [4] College of Pharmacy and Health Sciences, St. John's University, Queens, NY, USA. [5] These authors contributed equally: Yutian Zou, Shaoquan Zheng, Xinhua Xie. ✉email: chenz@stjohns.edu; tanghl@sysucc.org.cn; xiexm@sysucc.org.cn

According to the latest global statistics, breast cancer has surpassed lung cancer as the most pervasive malignancy worldwide, with an estimated 2.3 million new cases in 2020[1]. Amplification and/or overexpression of human epidermal growth factor receptor 2 (HER2) occurs in 14–30% of all breast cancer cases, which are defined as HER2-positive breast cancer; this subtype is associated with a worse survival outcome[2–4]. Trastuzumab (also named as heceptin), the first human monoclonal antibody against HER2, remarkably prolongs survival in patients with early-stage or metastatic HER2-positive breast cancer[5,6]. Despite the efficacy of trastuzumab, anti-HER2 resistance has emerged as a major cause of treatment failure in patients with HER2-positive breast cancer. Trastuzumab resistance is common in patients with HER2-positive breast cancer who receive adjuvant therapy, with a 1-year recurrence of nearly 15% and a 10-year recurrence over 31% after trastuzumab-based therapy in HERA trial[7]. The median progression-free survival for patients with advanced breast cancer treated with trastuzumab was ~11 months, and most patients developed secondary resistance within 1 year of receiving trastuzumab-based treatment[8]. Although a series of anti-HER2 drugs have been developed and entered clinical application, drug resistance still exists in a subset of patients. In APHINITY study, trastuzumab and pertuzumab dual anti-HER2 strategy had made great progress in curing early HER2-positive breast cancer, however, approximately one in ten patients still relapsed within 6 years[9]. Over 60% of patients with metastatic breast cancer still suffered progressed disease treated with combination of tucatinib (an anti-HER2 tyrosine kinase inhibitor) and trastuzumab in HER2CLIMB trial[10]. According to the result in EMILIA study, more than half of the patients did not respond to trastuzumab emtansine (TDM-1) as the second line of treatment for HER2-positive metastatic breast cancer[11]. It is still urgent to elucidate the underlying molecular mechanism of anti-HER2 resistance and develop new therapeutic strategies to overcome resistance.

Previous studies have uncovered several mechanisms of anti-HER2 resistance in HER2-positive breast cancer. First, the loss of trastuzumab binding site against the HER2 extracellular domain. For example, p95 HER2 truncated protein or MUC4 overexpression leads to the loss of the trastuzumab binding site and subsequent protection of HER2 from being blocked on the membrane, respectively[12,13]. Second, constitutive activation of the HER2 pathway is caused by dysregulation of downstream signaling components, including PIK3CA mutation or PTEN loss, in breast cancer cells[14–16]. Third, enhanced expression of receptor tyrosine kinases (RTKs) results in the compensatory activation of bypass signaling pathways. For instance, IGF1R expression was upregulated in anti-HER2 resistant breast cancer and maintained intracellular signaling flux after HER2 blockade[17,18]. Fourth, tumor immune infiltrates, including lymphocytes, dendritic cells, and natural killer cells, have also been reported to modulate responses and resistance to anti-HER2 therapy[19–21]. For example, endocytosis decreases the response to trastuzumab blockade via reducing the ADCC-mediating effect[22]. Despite the efforts of previous studies, the complete picture of the molecular mechanisms triggering anti-HER2 resistance in breast cancer remains unclear. Most inhibitors developed against these targets have failed to overcome anti-HER2 resistance in clinical trials. Additionally, there is a lack of biomarkers for accurately predicting the treatment response and recurrence risk in patients after neoadjuvant and adjuvant anti-HER2 therapy. Therefore, identifying robust drug resistance targets at the genome level is essential for providing new treatment options for patients who do not respond to anti-HER2 treatment.

In the present study, we conduct genome-scale CRISPR/Cas9 in vitro and in vivo screening to search for vulnerabilities of anti-HER2 resistant breast cancer. Containing 123,411 sgRNAs targeting 20,914 human genes, the GeCKOv2.0 human library is used as a functional profiling pool for CRISPR/Cas9 genetic selection, which led to the discovery of a series of genes involved in anti-HER2 resistance. Among these candidate genes and related proteins, fibroblast growth factor receptor 4 (FGFR4) ranks high in the results of both in vitro and in vivo screening. As an RTK, FGFR4 promotes metastasis, angiogenesis, chemoresistance, and the stemness of cancer cells in multiple digestive system neoplasms[23]. Although FGFR4 is one of the clustered genes included within the 50-gene intrinsic subtype predictor (PAM50)[24], little is known about the potential role of FGFR4 in breast cancer. Other FGFR family members have been reported as important mediators of anti-HER2 resistance. FGFR1 confers acquisition of resistance to lapatinib, trastuzumab, and TDM-1 in breast cancer[25,26]. FGFR2 signaling serves as an escape pathway which is responsible for anti-HER2 therapy resistance in breast cancer[27]. Autocrine loop driven by FGFR3 sustains acquired trastuzumab resistance in HER2-positive gastric cancer[28]. Nevertheless, there is a lack of study investigating the function of FGFR4 in anti-HER2 resistance. Here, we discover that FGFR4 expression is upregulated, which confers anti-HER2 resistance by attenuating ferroptosis in breast cancer. Patient-derived xenograft and organoid models reveal the potent efficacy of roblitinib (a selective inhibitor of FGFR4; also named as FGF-401) in both intrinsic and acquired anti-HER2 resistant breast cancer. Thus, this study pinpoints a mechanism of anti-HER2 resistance and provides a strategy for overcoming this resistance by inhibiting FGFR4 in HER2-positive breast cancer.

## Results

**Genome-wide CRISPR screening identifies FGFR4 as a crucial gene for anti-HER2 resistance in breast cancer.** Trastuzumab-based anti-HER2 regimen is the current standard for HER2-positive breast cancer therapy. Therefore, we generated anti-HER2 resistant cells by continuously exposing trastuzumab-sensitive HER2-positive breast cancer cell lines SKBR3, BT474, and AU565 to trastuzumab for 3 months in vitro (Supplementary Fig. 1a). The anti-HER2 resistant cell lines rSKBR3, rBT474, and rAU565 showed higher IC50 values for trastuzumab and undisturbed proliferation than those for their respective parental cell lines (Supplementary Fig. 1b). Bright-field micrographs of cultured parental cells and resistant cells treated with vehicle or trastuzumab are showed in Supplementary Fig. 1c. The generated rSKBR3, rBT474, and rAU565 cells remained stably anti-HER2 resistance after withdrawing from trastuzumab for 4 weeks (Supplementary Fig. 1d). In addition, in vivo experiments confirmed that resistant cell lines had significant tolerance to trastuzumab treatment (Supplementary Fig. 1e–g). To identify the vulnerabilities of trastuzumab-resistant breast cancer cells, we conducted a genome-wide CRISPR screening on rSKBR3 cells with an sgRNA lentiviral library containing 123,411 sgRNAs targeting 20,914 human genes (Fig. 1a). To ensure the accuracy of loss-of-function genetic screening, both in vitro and in vivo selection were conducted. In this screening strategy, cells carrying sgRNA targeting genes that are crucial for viability would be depleted under trastuzumab treatment conditions (i.e., resistance genes). After high-throughput screening, 1052 genes and 1032 genes were identified after in vitro and in vivo selection, respectively. Among them, several identified genes (IGF1R[18], SRC[29], PIK3CA[16], CTNNB1[30], CCNE1[31], FOXM1[32], CDK12[33], etc.) have been reported as robust anti-HER2 resistant genes in previous studies, confirming the fitness of our screening approach. Functional and pathway enrichment analyses of these resistance genes were further conducted in Metascape, among which FGFR

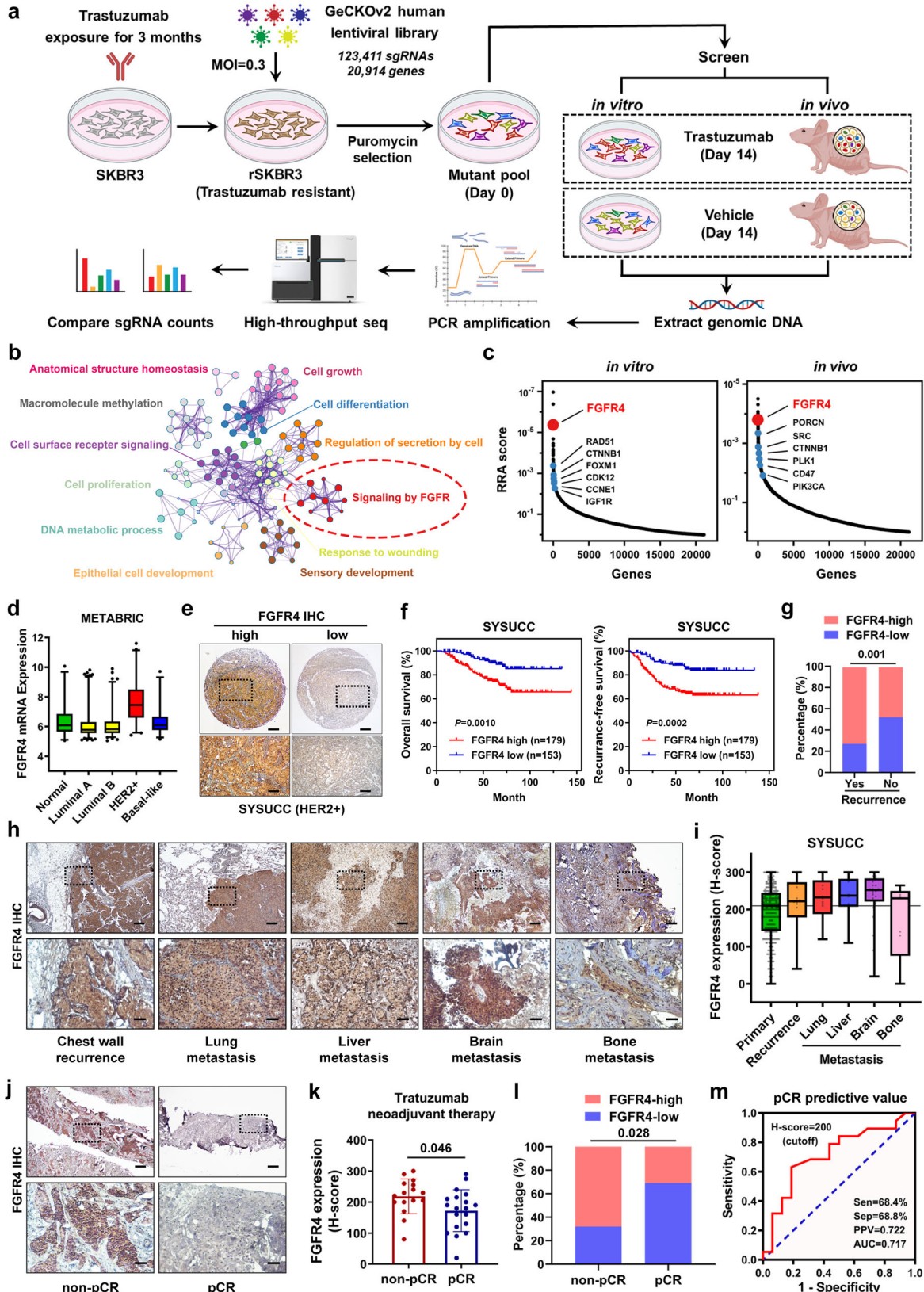

signaling was one of the most important pathways involved in anti-HER2 resistance (Fig. 1b). As one of the core family members of the FGFR signaling pathway, *FGFR4* ranked high in both the in vitro and in vivo screens (Fig. 1c). FGFR4 is a druggable target; the first-in-class highly selective and potent inhibitor developed for this protein is roblitinib (FGF-401)[34]. Roblitinib

demonstrated promising clinical efficacy and a favorable safety profile in phase II clinical trials for the treatment of hepatocellular carcinoma and solid tumors with high FGFR4 expression[35]. Therefore, we chose FGFR4 for further investigation to validate whether it could be a target to overcome anti-HER2 resistance in breast cancer. According to the TCGA and METABRIC

**Fig. 1 Genome-wide CRISPR screening identifies *FGFR4* as a crucial gene for anti-HER2 resistance in breast cancer. a** Flowchart of genome-wide CRISPR screening of anti-HER2 resistance-associated genes using the pooled human GeCKOv2.0 lentivirus sgRNA library. **b** Functional and pathway enrichment analysis of anti-HER2 resistance-associated genes as conducted with Metascape. **c** The robust ranking aggregation (RRA) score reveals context-dependent vulnerabilities in anti-HER2 resistant breast cancer. The blue dots represent anti-HER2 resistant genes that have been reported by previous studies. **d** *FGFR4* was highly expressed in HER2-positive breast cancer in the METABRIC database. Normal like ($n = 140$), Luminal A ($n = 679$), Luminal B ($n = 461$), HER2-positive ($n = 220$), Basal-like ($n = 199$). **e** Representative IHC staining images showing high or low expression of FGFR4 in 322 HER2-positive breast cancer tissues from SYSUCC. Scale bar = 200 µm (low magnification) and 100 µm (high magnification). **f** Kaplan-Meier analysis of overall and recurrence-free survival of HER2-positive breast cancer patients in the SYSUCC cohort with high or low FGFR4 expression. **g** The expression of FGFR4 by IHC H-score in specimens of HER2-positive breast cancer from patients with ($n = 80$) or without ($n = 252$) recurrence. **h, i** Representative images and H-score of FGFR4 expression in different metastases of breast cancer after IHC staining. The samples include primary ($n = 332$), recurrence ($n = 16$), lung ($n = 16$), liver ($n = 8$), brain ($n = 28$), and bone metastasis ($n = 9$). Scale bar = 200 µm (low magnification) and 100 µm (high magnification). **j** Representative images of FGFR4 IHC staining in non-pCR and pCR breast cancer tissues after anti-HER2-based neoadjuvant therapy. Samples were collected by core-needle biopsy prior to therapy. Scale bar = 200 µm (low magnification) and 100 µm (high magnification). **k** IHC H-score of FGFR4 in non-pCR ($n = 16$) and pCR ($n = 19$) breast cancer samples. Data were presented as mean ± S.D. **l** The expression of FGFR4 in specimens of HER2-positive breast cancer from patients with ($n = 19$) or without pCR ($n = 16$). **m** Receiver operating characteristic curve depicting the accuracy of FGFR4 expression in predicting pCR during anti-HER2-based neoadjuvant therapy. Data were analyzed by log-rank test in **f**, chi-square test in **g, l**, two-sided Student's *t* test in **k**. Source data are provided as a Source Data file.

databases, *FGFR4* mRNA was highly overexpressed in HER2-positive breast cancer (Fig. 1d and Supplementary Fig. 2a, b). FGFR4 was highly expressed in recurrence group (FinHER trial[36]) and non-pathological complete response (non-pCR) group (CHER-LOB trial[37]) after anti-HER2 adjuvant and neoadjuvant therapy, respectively (Supplementary Fig. 2c). FGFR4 expression was detected in 332 cases of HER2-positive breast cancer receiving adjuvant anti-HER2 treatment in the SYSUCC cohort (Fig. 1e). High expression of FGFR4 was associated with worse recurrence-free survival and overall survival in SYSUCC cohort (Fig. 1f). Patients with high FGFR4 tumors had a higher rate of recurrence after surgery (Fig. 1g and Supplementary Fig. 2d, e). Further univariate and multivariate Cox regression analyses revealed that FGFR4 expression was an independent risk factor for HER2-positive breast cancer patients (Supplementary Tables 1–3). To verify the expression level of FGFR4 in advanced breast cancer, we collected specimens from different sites of recurrence or metastasis. Compared to primary breast cancers, metastatic tumors showed higher FGFR4 expression, especially brain metastases of breast cancer (Fig. 1h, i and Supplementary Fig. 2f). To examine whether FGFR4 expression correlates with intrinsic anti-HER2 resistance, we performed IHC staining using 36 HER2-positive breast cancer biopsies obtained before administration of anti-HER2 based neoadjuvant therapy (Fig. 1j). Patients with non-pCR tumors had higher FGFR4 expression than those with pCR tumors (Fig. 1k, l). Additionally, a receiver operating characteristic curve indicated that FGFR4 expression could be used as a predictor of pCR before neoadjuvant anti-HER2 therapy (Fig. 1m).

**FGFR4 inhibition enhances sensitivity to anti-HER2 treatment in both intrinsic and acquired resistant HER2-positive breast cancer cell lines.** *FGFR4* mRNA was expressed at low levels in normal mammary epithelial cells (MCF-10A) and triple-negative breast cancer cell lines (Fig. 2a). Compared to anti-HER2 sensitive HER2-positive cell lines (SKBR3, BT474, and AU565), intrinsically anti-HER2 resistant HER2-positive cell lines (MDA-MB-453 and MDA-MB-361) had higher *FGFR4* expression level (Fig. 2a). Consistently, *FGFR4* mRNA levels were significantly higher in acquired anti-HER2 resistant HER2-positive cell lines (rSKBR3, rBT474, and rAU565) than in their corresponding parental cell lines (Fig. 2b). According to the Cancer Therapeutics Response Portal (CTRP) database, cells with resistance to lapatinib (an anti-HER2 inhibitor) had higher levels of *FGFR4* expression (Fig. 2c). To explore the effect of FGFR4 in driving anti-HER2 resistance, we established FGFR4 overexpression

models in anti-HER2 sensitive breast cancer cells (Fig. 2d). To better simulate the effect of small-molecule inhibitors, the dCas9-KRAB CRISPRi technique was used to suppress FGFR4 expression in anti-HER2 resistant cells (Fig. 2e). Transcriptional repression of CRISPRi had an extremely high efficiency in inhibiting FGFR4 expression, as validated by qPCR and immunofluorescence analysis (Fig. 2f, g). As the foundation of anti-HER2 regime, trastuzumab sensitivity was evaluated after FGFR4 intervention. Trastuzumab resistance was significantly reversed after inhibiting FGFR4 in cells with either intrinsic (MDA-MB-453 and MDA-MB-361) or acquired (rSKBR3) resistance (Fig. 2h and Supplementary Fig. 3a). In addition, exogenous overexpression of FGFR4 conferred resistance to trastuzumab in SKBR3 and BT474 cell lines (Fig. 2i). Roblitinib, an FGFR4 selective inhibitor, was used for further evaluation (Supplementary Fig. 3b). Roblitinib exhibited a potent antitumor effect on trastuzumab-resistant cells (Fig. 2j and Supplementary Fig. 3c). Additionally, the combination of trastuzumab and roblitinib showed synergistic effect in trastuzumab-resistant cells with a combination index lower than value 1 (Fig. 2j). Real-time monitoring further confirmed the combined effect of trastuzumab and roblitinib in treating trastuzumab-resistant HER2-positive breast cancer cells (Fig. 2k and Supplementary Fig. 3d). Cotreatment with trastuzumab and roblitinib led to a remarkable reduction in colony formation in cells with either intrinsic or acquired resistance in a concentration-dependent manner (Fig. 2l). The combination reduced the property of stem cells, the ability to form mammospheres, suggesting it reduces stem cells, which have been implicated in recurrence/metastasis. (Fig. 2m and Supplementary Fig. 3e). Next, we further investigated the effect of FGFR4 on the sensitivity of other anti-HER2 agents currently employed in the clinic. Trastuzumab emtansine (TDM-1) is an antibody-drug conjugate approved as the second line of treatment for HER2-positive metastatic breast cancer[11]. Exogenous overexpression of FGFR4 reduced the sensitivity to TDM-1 in SKBR3 cell lines, while FGFR4 suppression increased the sensitivity to TDM-1 treatment in rSKBR3 cell lines (Supplementary Fig. 3f). Trastuzumab plus pertuzumab strategy is the current standard-of-care for adjuvant[38], neoadjuvant[39], and metastatic[40] treatment in HER2-positive breast cancer. As a HER2-targeted tyrosine kinase inhibitor, tucatinib is used with trastuzumab in metastatic HER2-positive breast cancer[10]. We found that FGFR4 reduced the sensitivity of HER2-positive breast cancer to trastuzumab plus pertuzumab or tucatinib (Supplementary Fig. 3g, h). Therefore, inhibition of FGFR4 might be a strategy for increasing sensitivity to multiple anti-HER2 strategies in breast cancer.

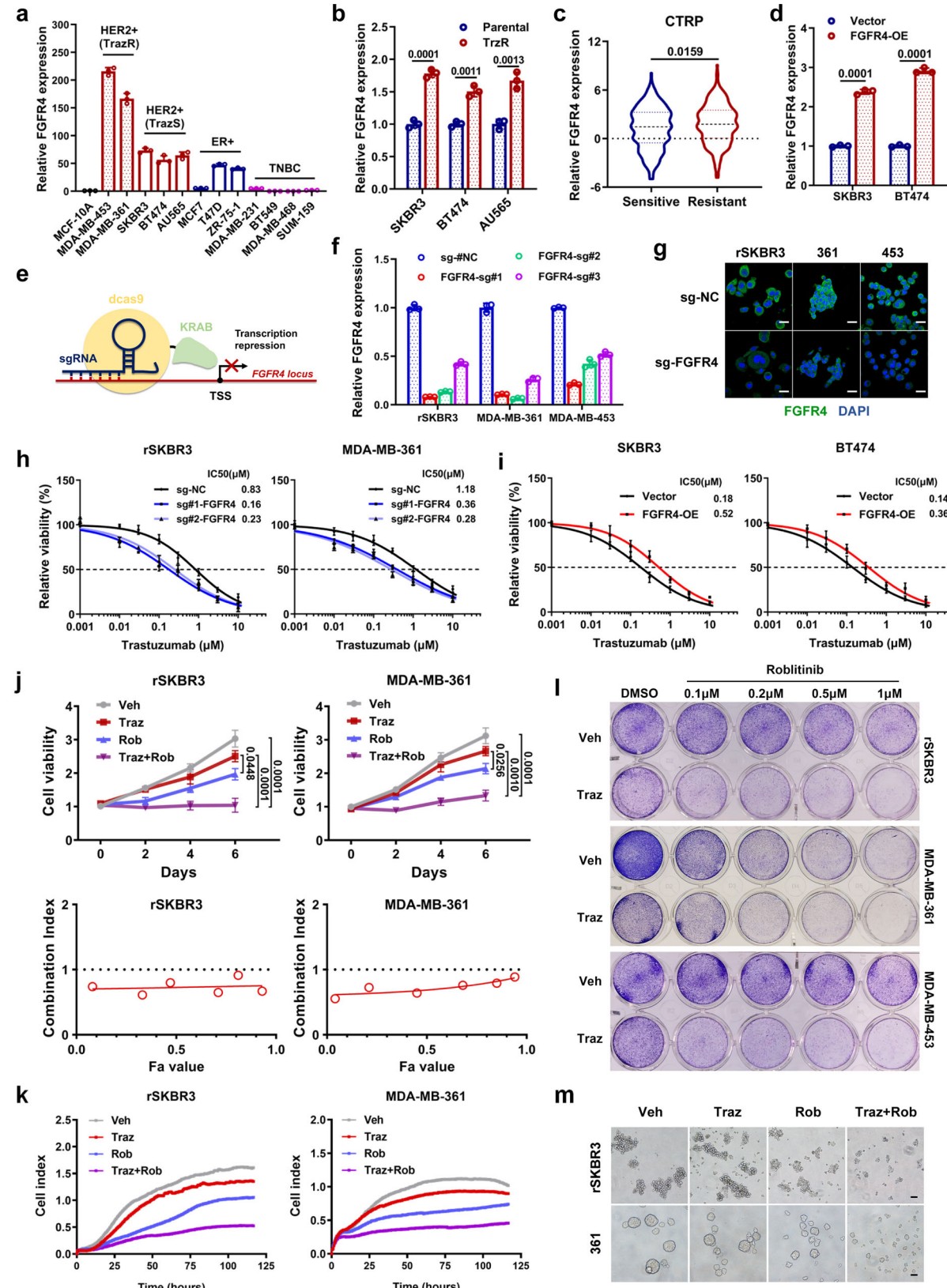

**m6A hypomethylation mediates FGFR4 upregulation in anti-HER2 resistant breast cancer**. To investigate the regulatory mechanism underlying *FGFR4* upregulation, *FGFR4* mRNA levels were examined to determine any pre- and posttranscriptional regulatory mechanisms. We found that *FGFR4* mRNA expression remained unchanged after treatment with 5-azacytidine (DNA

methylation inhibitor) or trichostatin A (HDAC inhibitor), indicating that neither DNA methylation nor histone acetylation is the reason for *FGFR4* upregulation (Supplementary Fig. 4a). N6-methyladenosine (m6A) modification is an important post-transcriptional regulatory mechanism of intracellular mRNA expression[41,42]. As detected by dot blot assays, the global m6A

**Fig. 2 FGFR4 inhibition enhances sensitivity to anti-HER2 treatment in both intrinsic and acquired resistant HER2-positive breast cancer cell lines.**
**a** The expression of *FGFR4* was detected in normal mammary cell lines and breast cancer cell lines with different subtypes by qPCR. **b** *FGFR4* expression in parental (SKBR3, BT474, and AU565) and trastuzumab-resistant (rSKBR3, rBT474, and rAU565) cell lines. **c** Cells with resistance to lapatinib (an anti-HER2 inhibitor) had higher levels of *FGFR4* expression, according to the CTRP database. Sensitive $n = 305$. Resistant $n = 298$. **d** The efficiency of *FGFR4* overexpression was examined by qPCR. **e** Schematic diagram of the CRISPRi method used to inhibit FGFR4 expression. **f** The inhibition efficiency of the three most effective sgRNAs as validated by qPCR analysis. **g** FGFR4 protein was detected after transcriptional repression and observed under a confocal microscope. Scale bar = 20 μm. **h** Dose–response curves of trastuzumab-resistant cells (rSKBR3 and MDA-MB-361) carrying sg-NC or sg-FGFR4 constructs after treatment with trastuzumab. Cell viability was measured using absorbance value at 450 nm, determined by CCK-8 assays. Group treated with vehicle (control) was defined as 100% relative viability. **i** Dose–response curves of trastuzumab-sensitive cells (SKBR3 and BT474) carrying empty vector or FGFR4 overexpression vector constructs after treatment with trastuzumab. **j** Cell viability was measured in trastuzumab-resistant cells treated with vehicle (Veh), 0.5 μM trastuzumab (Traz) and/or 0.5 μM roblitinib (Rob, FGFR4 inhibitor). Synergistic effect of anti-FGFR4 with anti-HER2 therapy was evaluated by Combination Index. **k** Real-time monitoring of live cells was performed to evaluate the efficacy of trastuzumab and roblitinib in trastuzumab-resistant cells. **l** Colony formation assays of rSKBR3, MDA-MB-361, and MDA-MB-453 cells treated with vehicle or trastuzumab (0.5 μM) with increasing concentrations of roblitinib. **m** Mammosphere formation assay revealed the effects of reagents on inhibiting HER2-positive breast cancer stem cells. Scale bar = 100 μm. Data in **a**, **b**, **d**, **h**, **i**, and **j** are presented as mean ± S.D., $n = 3$ biologically independent samples. Data were analyzed by two-sided Student's *t* test in **b**–**d**, and one-way ANOVA adjusted for multiple comparisons for **j**. Source data are provided as a Source Data file.

level was extremely decreased in trastuzumab-resistant cells compared to trastuzumab-sensitive cells (Fig. 3a). Immunofluorescence imaging and MeRIP assays also revealed a reduction in m6A levels in resistant cells, indicating that a decline in m6A levels may play an important role in FGFR4 upregulation (Fig. 3b, c and Supplementary Fig. 4b). The expression of m6A writers (methylase) and erasers (demethylase) was evaluated in resistant and parental HER2-positive breast cancer cells. Among these candidates, *METTL14* showed remarkably decreased expression in trastuzumab-resistant cells, which could be responsible for the reduction in m6A levels (Fig. 3d). Western blot assays also confirmed that METTL14 expression was decreased in trastuzumab-resistant breast cancer cells (Supplementary Fig. 4d). Therefore, we speculate that METTL14 reduction causes the upregulation of *FGFR4* expression in HER2-positive breast cancer. Low *METTL14* expression was associated with worse recurrence-free survival in HER2-positive breast cancer (Fig. 3e). Knockdown of METTL14 significantly increased the expression level of *FGFR4* in HER2-positive breast cancer cells (Fig. 3f). *FGFR4* expression was negatively correlated with *METTL14* in the TCGA database (Supplementary Fig. 4c). To further confirm the exact m6A modification sites on *FGFR4* mRNA, we used the m6A-Atlas database to predict m6A motifs based on several MeRIP-seq results. Luciferase reporter assays revealed that activity was increased in the wild-type *FGFR4* mRNA group after silencing METTL14, but no changes were observed in the mutant group (Fig. 3g). In addition, two m6A modification sites were validated via step-by-step mutation of the reporting plasmid (Fig. 3k). MeRIP assays further confirmed the role of METTL14 in m6A modification of *FGFR4* mRNA (Fig. 3h, i). Furthermore, *FGFR4* mRNA stability was increased in METTL14-deficient cells, as evaluated by the actinomycin D assay (Fig. 3j). Since the decay of *FGFR4* mRNA was influenced by m6A modification, we next examined the impact of six stability-related m6A readers (IGF2BP1, IGF2BP2, IGF2BP3, YTHDC2, YTHDF2, and YTHDF3) on *FGFR4* mRNA expression. As shown in Fig. 3l, only YTHDC2 influenced *FGFR4* expression in rSKBR3 cells. A negative correlation between *FGFR4* expression and *YTHDC2* levels was observed in the TCGA cohort of HER2-positive breast cancer patients (Fig. 3m). Low expression of *YTHDC2* was associated with worse recurrence-free survival in patients with HER2-positive breast cancer (Fig. 3n). The direct interaction of the YTHDC2 protein with *FGFR4* mRNA was confirmed by RNA immunoprecipitation assays (Fig. 3o and Supplementary Fig. 4e). Luciferase reporter assays revealed that the luminescence intensity was increased in the wild-type group but not in the mutant group after YTHDC2 was silenced (Fig. 3p). The *FGFR4* mRNA

decay rate was significantly decreased after knockdown of YTHDC2 in MDA-MB-361 cells (Fig. 3q). To accurately modify RNA with m6A, we introduced the dCas13b-METTL3/14 fusion system with sgRNAs targeting *FGFR4* mRNA for further validation (Fig. 3r). Two sites were targeted and evaluated via MeRIP-PCR analysis (Fig. 3s, t). *FGFR4* mRNA levels were significantly reduced after m6A modification in rSKBR3 and MDA-MB-361 cells (Supplementary Fig. 4f). Sensitivity to trastuzumab was restored after m6A modification (Fig. 3u). We next explored the mutations in the trastuzumab-resistant cell line which may explain METTL14 reduction or FGFR4 overexpression. We compared the gene alterations between resistance and parental SKBR3 breast cancer cells. In total, 38 altered genes were identified with high biological impact in the trastuzumab-resistant cell line (Supplementary Table 4). However, these 38 mutations are rarely found in HER2-positive breast cancer according to TCGA and METABRIC databases (Supplementary Fig. 4g, h). Additionally, these 38 mutations were not correlated with the METTL14 reduction or FGFR4 overexpression in TCGA and METABRIC database (Supplementary Fig. 4i). These mutations were also not found in the resistant breast cancer patients recruited in previous clinical trials (CALGB-40601[43] and CHER-LOB study[37]) (Supplementary Fig. 4i). These results indicated that the METTL14 reduction or FGFR4 overexpression was less likely to be caused by genomic alteration; however, more likely due to epigenetic change and transcriptome adaptation. Together, our data demonstrated that m6A RNA hypomethylation resulted in FGFR4 upregulation, which confers anti-HER2 resistance to breast cancer.

**FGFR4 phosphorylates GSK-3β to modulate β-catenin/TCF signaling and drive anti-HER2 resistance.** To elucidate how FGFR4 confers anti-HER2 resistance in breast cancer, we performed RNA next-generation sequencing in FGFR4-suppressed and control rSKBR3 cells (Fig. 4a). Overall, 327 genes had downregulated expression whereas 374 genes had upregulated expression based on fold change > 2 and FDR < 0.05, as shown in the volcano plot (Fig. 4b). Differentially expressed genes were further analyzed by KEGG enrichment analysis. The results revealed that the WNT signaling pathway was an important downstream target of FGFR4 signaling in resistant HER2-positive breast cancer cells (Fig. 4c). GSEA also showed a strong correlation between FGFR4 and the WNT signaling pathway (Fig. 4d). According to previous studies, activation of WNT/β-catenin signaling cascade remarkably drives anti-HER2 resistance in breast cancer[30,33,44]. FGFR4 can directly phosphorylate GSK-3β to stimulate WNT/β-catenin signaling in several malignancies[23].

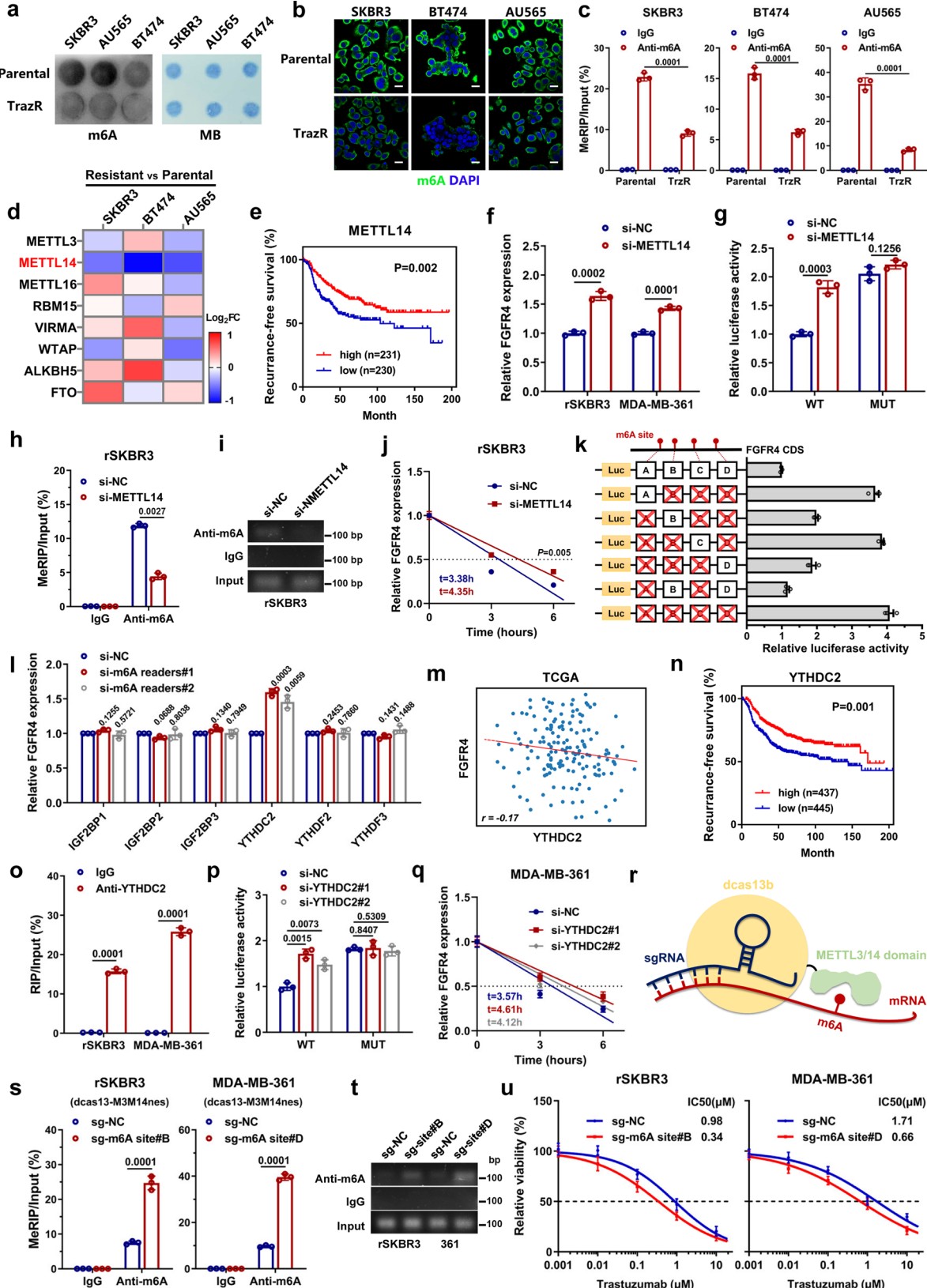

We found that phosphorylated GSK-3β was increased and WNT/β-catenin signaling was activated in trastuzumab-resistant breast cancer cell lines (Supplementary Fig. 5a). In addition, resistant breast cancer cells gradually reduced their dependence on MAPK and PI3K/AKT signaling which are vital to sensitive parental breast cancer cells[45,46] (Supplementary Fig. 5b). Therefore, we

assumed that FGFR4 mediates anti-HER2 resistance via activating WNT signaling pathway. Inconsistent with the hypothesis, knockdown of FGFR4 significantly reduced the phosphorylation of GSK-3β and inhibited the WNT signaling pathway via downregulation of active β-catenin (Fig. 4e). The WNT signaling pathway was also blocked after treatment with the

**Fig. 3 m6A hypomethylation mediates FGFR4 upregulation in anti-HER2 resistant breast cancer.** Global m6A levels were measured in parental and resistant cells using **a** m6A dot blot assays, **b** immunofluorescent staining, and **c** MeRIP-PCR analysis. Scale bar = 10 μm. **d** The expression of eight m6A writers/erasers was detected by qPCR in parental and resistant cells. **e** Recurrence-free survival of HER2-positive breast cancer patients with high ($n = 231$) or low ($n = 230$) *METTL14* expression in KM-Plotter database. **f** *FGFR4* expression was enhanced after METTL14 silencing. **g** Luciferase reporter assays revealed that activity was increased in the wild-type *FGFR4* mRNA group after silencing METTL14, but not changed after the m6A sites were mutated. **h** MeRIP-qPCR analysis of *FGFR4* mRNA enrichment after METTL14 knockdown. **i** Agarose electrophoresis analysis of *FGFR4* mRNA PCR products after MeRIP assays. **j** The degradation rate of *FGFR4* mRNA after actinomycin D exposure. **k** Two m6A modification sites were validated through step-by-step mutation of luciferase reporter. **l** *FGFR4* expression was detected after multiply silencing different m6A readers. **m** Correlation between *FGFR4* and *YTHDC2* expression in HER2-positive breast cancer specimens in TCGA database. Spearman test was used. **n** Recurrence-free survival of HER2-positive breast cancer patients with high ($n = 437$) or low ($n = 445$) *YTHDC2* expression in KM-Plotter database. **o** qPCR analysis of RIP assays in breast cancer cells confirmed the direct binding between the YTHDC2 protein and *FGFR4* mRNA. **p** Luciferase reporter assays revealed that activity was increased in the wild-type *FGFR4* mRNA group after silencing YTHDC2. **q** The degradation rate of *FGFR4* mRNA after actinomycin D exposure. **r** Schematic diagram of precise m6A modification of the dCas13b-METTL3/14 fusion system. **s, t** qPCR and agarose electrophoresis analysis of *FGFR4* mRNA enrichment after MeRIP assays. **u** Dose–response curves of cells carrying negative control or sgRNA constructs after treatment with trastuzumab. Data in **c**, **f**, **g**, **h**, **j**, **k**, **l**, **p**, **q**, **s**, and **u** were presented as mean ± S.D., $n = 3$ biologically independent samples. Data were analyzed by two-sided Student's *t* test in **c**, **f-h**, **j**, **l**, **o**, **p**, **s**, and log-rank test for **e**, **n**. Source data are provided as a Source Data file.

FGFR4 inhibitor roblitinib in HER2-positive trastuzumab-resistant cells (Fig. 4f). Additionally, overexpression of FGFR4 resulted in the elevation of phosphorylated GSK-3β and active β-catenin levels (Fig. 4g). Accumulation of β-catenin in the nucleus is the marker of WNT pathway activation. Nuclear β-catenin expression was increased or decreased after FGFR4 knockdown or overexpression, respectively (Fig. 4h). Immunocytochemistry and immunofluorescence staining clearly revealed the levels of β-catenin in the nucleus, which were significantly reduced after FGFR4 inhibition (Fig. 4i–l). Knockdown of FGFR4 showed limited effect on the canonical MAPK and PI3K/AKT pathway in trastuzumab-resistant cells, possibly due to their reduced dependence on these two signaling pathways (Supplementary Fig. 5c). Then, dCas13b-METTL3/14 fusion system was used as described above to confirm the impact of *FGFR4* mRNA m6A modification on WNT/β-catenin signaling pathway. Western blot assays revealed that WNT/β-catenin signaling was attenuated after increasing the m6A-modified level of *FGFR4* mRNA in rSKBR3 breast cancer cells (Supplementary Fig. 5d). Furthermore, inhibition of WNT/β-catenin signaling restored the sensitivity to trastuzumab treatment in rSKBR3 and MDA-MB-361 breast cancer cells (Supplementary Fig. 5e, f).

**Ferroptosis is triggered following FGFR4 inhibition in anti-HER2 resistant breast cancer cells.** To further clarify the biological role of FGFR4 in regulating anti-HER2 resistance, GSEA was subsequently conducted. Interestingly, we found that FGFR4 was greatly associated with glutathione metabolism and iron ion homeostasis, which are two core pathways of ferroptosis (Fig. 5a). Therefore, we speculated that FGFR4 inhibition overcomes anti-HER2 resistance by triggering ferroptosis in breast cancer. A series of experiments was conducted to evaluate glutathione content and lipid peroxidation upon FGFR4 knockdown. The ratio of glutathione to oxidized glutathione was decreased after inhibiting FGFR4 (Fig. 5b and Supplementary Fig. 6a). Liperfluo staining visualized lipid ROS, the amount of which was increased after FGFR4 suppression (Fig. 5c–f). Moreover, lipid ROS gradually accumulated on the cytomembrane after FGFR4 inhibition, as revealed by flow cytometry at different time periods after this inhibition (Fig. 5g). C11-BODIPY probe staining shows that the ratio of oxidized to nonoxidized lipids was remarkably increased after FGFR4 inhibition and reversed by ferroptosis specific inhibitor ferrostatin-1, as assessed in by two-channel flow cytometry analysis (Fig. 5h). Confocal microscopy visualized the alterations in lipid peroxidation in rSKBR3 and MDA-MB-361 cells (Fig. 5i and Supplementary Fig. 6j). Additionally, the level of

lipid peroxidation product (MDA) was significantly increased in FGFR4-deficient cells (Fig. 5j and Supplementary Fig. 6b). Transmission electron microscopy revealed a distinctive morphological feature of ferroptosis in cells subject to FGFR4 inhibition (Fig. 5k). Mitochondria appeared smaller than normal with increased membrane density, consistent with the description in previous studies[47,48]. Next, we measured the level of intracellular ferrous ions ($Fe^{2+}$) by using FerroOrange probes and colorimetric iron assays. The labile iron pool increased after treatment with the FGFR4 inhibitor, which confirmed the function of FGFR4 in iron homeostasis (Fig. 5l, m and Supplementary Fig. 6c). Cell death caused by roblitinib was mainly rescued by cotreatment with either ferrostatin-1, liproxstatin-1, or the iron chelator deferoxamine, providing evidence for the ferroptotic nature of roblitinib (Fig. 5n, o). Lactate dehydrogenase (LDH) release assays revealed that cell death caused by FGFR4 inhibition could be reversed by ferrostatin-1 (Supplementary Fig. 6d). Next, we performed flow cytometry assays after liperfluo and sytox staining to detect oxidized and dead cells, respectively. Liperfluo and sytox positivity rate was increased after roblitinib treatment and rescued by the presence of ferrostatin-1 (Supplementary Fig. 6e–g). To further identify the type of cell death induced by FGFR4 inhibition, flow cytometry assay was conducted after Annexin V/PI staining. Roblitinib-induced cell death was overwhelmingly rescued by ferrostatin-1 and slightly rescued by Z-DEVD-FMK (a caspase inhibitor) (Supplementary Fig. 6h, i). Western blot assay showed cleaved caspase-3 was increased after roblitinib treatment and reduced by Z-DEVD-FMK, which indicated apoptosis was also induced although the effect was not obvious (Supplementary Fig. 6o). Time-lapse movie recorded the morphological changes before cell death after FGFR4 inhibition in rSKBR3 and MDA-MB-453 resistant breast cancer cells (Supplementary Movies 1 and 2). Nuclei and cytomembrane are intact before cell death which is the characteristic of ferroptosis (Supplementary Fig. 6k). Additionally, we also visualized the lipid peroxide generation and cell death process by time-lapse fluorescence imaging (Supplementary Movies 3 and 4). Lipid peroxide was accumulated before cell death in rSKBR3 and MDA-MB-453 cells treated with roblitinib (Supplementary Fig. 6l, m). To investigate the impact of ferroptosis on trastuzumab resistance, four different classes of ferroptosis inducers were used (Erastin, FIN56, FINO2, and RSL-3). Directly triggering of ferroptosis significantly restored trastuzumab sensitivity in rSKBR3 breast cancer cells (Supplementary Fig. 6n). These results indicated ferroptosis was the main form of cell death after FGFR4 inhibition which increased the sensitivity of anti-HER2 treatment in breast cancer.

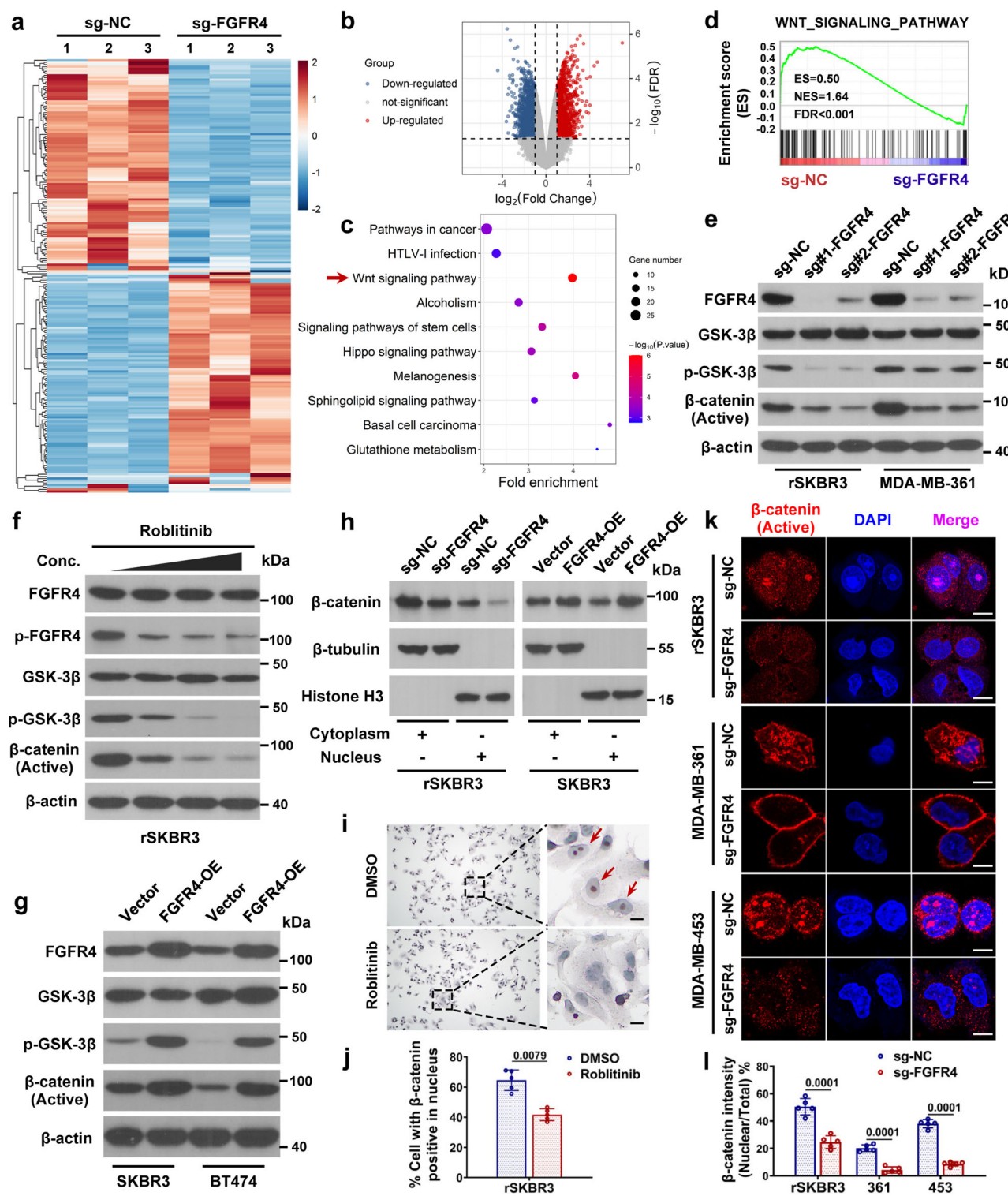

**FGFR4 accelerates cystine uptake and Fe²⁺ efflux via the β-catenin/TCF4-SLC7A11/FPN1 axis**. To gain insight into the specific mechanism by which FGFR4 regulates glutathione metabolism and iron ion homeostasis, we analyzed the gene expression of these two pathways. We found that after FGFR4 inhibition, the cystine transporter encoding gene *SLC7A11* and iron transporter encoding gene *FPN1* (official name as *SLC40A1*) was one of the most downregulated genes in glutathione metabolism and iron ion homeostasis gene set, respectively (Fig. 6a). This result was validated by qPCR and immunofluorescence

analysis (Fig. 6b, c). *SLC7A11* and *FPN1* expression was upregulated in trastuzumab-resistant HER2-positive breast cancer cells compared to parental cells (Fig. 6d). Since FGFR4 promotes anti-HER2 resistance by activating the downstream β-catenin pathway, we hypothesized that the upregulation of *SLC7A11* and *FPN1* expression is mediated by the transcriptional effect of the β-catenin/TCF-4 complex. We analyzed the ChIP-seq results of TCF-4 protein in the ENCODE public database and found several binding peaks of TCF-4 in the promoter regions of the *SLC7A11* and *FPN1* loci (Fig. 6e, f and Supplementary Fig. 7a–c). Several

**Fig. 4 FGFR4 phosphorylates GSK-3β to modulate β-catenin/TCF signaling and drive anti-HER2 resistance. a** Heatmap indicating the top 100 upregulated and downregulated genes between sg-NC (negative control) and sg-FGFR4 rSKBR3 cells as detected by RNA next-generation sequencing with three biological duplicates. **b** Volcano plot showing differentially expressed genes based on absolute fold change >2 and FDR < 0.05. **c** Differentially expressed genes were analyzed by Kyoto Encyclopedia of Genes and Genomes (KEGG) enrichment analysis. *P* values were calculated by hypergeometric test and adjusted for multiple comparisons. **d** Gene set enrichment analysis (GSEA) shows a strong correlation between FGFR4 and the WNT signaling pathway. **e** Western blot assays showing that the levels of the indicated proteins were altered after FGFR4 inhibition in resistant HER2-positive breast cancer cells. **f** Effect of the FGFR4 inhibitor roblitinib on the WNT signaling pathway in rSKBR3 breast cancer cells as assessed by western blot analysis. **g** Overexpression of FGFR4 resulting in the elevation of phosphorylated GSK-3β levels and active β-catenin levels in trastuzumab-sensitive SKBR3 and BT474 cells. **h** Subcellular localization of β-catenin as analyzed by western blot after cellular fractionation. **i, j** Immunocytochemistry staining showing a reduction in β-catenin accumulation in the nucleus after treatment with the FGFR4 inhibitor in rSKBR3 cells. Scale bar = 10 μm. **k, l** Laser confocal imaging of immunofluorescence staining analysis clearly shows that the level of β-catenin in the nucleus was significantly reduced after FGFR4 inhibition. Scale bar = 10 μm. Data in **j** and **l** were presented as mean ± S.D., *n* = 5 biologically independent samples. Data were analyzed by two-sided Student's *t* test in **j** and **l**. Source data are provided as a Source Data file.

binding elements and motifs were identified by the JASPAR website (Fig. 6h). Then, ChIP assays were conducted to evaluate the potential TCF-4 binding sites in the *SLC7A11* and *FPN1* promoter regions (Fig. 6g). The results showed that *SLC7A11* promoter region #3 (−1500 to −1250 before the TSS) and region #8 (−250 to 0 before the TSS) were the most abundant among regions enriched by the TCF-4 antibody (Fig. 6i, j). For the *FPN1* promoter, region #3 (−1500 to −1250 before the TSS) and region #6 (−750 to −500 before the TSS) were enriched after ChIP assays (Fig. 6i, j). Inhibition of FGFR4 significantly reduced the recruitment of the β-catenin/TCF-4 complex to both the *SLC7A11* and *FPN1* promoters (Fig. 6k, l). To further validate the specific interaction sites, we constructed a series of luciferase reporter vectors carrying the wild-type or mutant *SLC7A11* or *FPN1* promoter regions. The results showed that the deletion of exact binding elements significantly reduced the relative luciferase activity (Fig. 6m, n). To validate the expression of SLC7A11 and FPN1 in human cohorts, 332 HER2-positive breast cancer samples from SYSUCC were retrospectively analyzed by IHC staining (Fig. 6o, p). A strong positive correlation between FGFR4 expression and SLC7A11 or FPN1 expression was discovered in HER2-positive breast cancer (Fig. 6q, r and Supplementary Fig. 7d). Additionally, high expression of SLC7A11 or FPN1 was correlated with poor recurrence-free survival in the SYSUCC cohort (Fig. 6s, t). IHC staining has been confirmed to be specific by negative controls in HER2-positive breast cancer cells (Supplementary Fig. 7e). Western blot analysis revealed that SLC7A11 and FPN1 expression was remarkably decreased after FGFR4 inhibition, and these effects could be rescued by over-expression of β-catenin (Fig. 6u). Alterations in the expression of GPX4, another marker of ferroptosis, were also observed. The expression of two other important proteins in the regulation of ferroptosis, the iron ion imports protein TFRC and the fatty acid metabolic protein ACSL4, remained unchanged after FGFR4 or β-catenin silence (Fig. 6u).

**Patient-derived models revealed the efficacy of FGFR4 inhibitor in both intrinsic and acquired anti-HER2 resistant breast cancers.** The antitumor efficacy of the FGFR4 inhibitor roblitinib was further validated in both intrinsic and acquired trastuzumab-resistant breast cancer models. First, rSKBR3 and MDA-MB-361 tumor-bearing mice were treated with vehicle, trastuzumab, roblitinib, or the combined agents (Fig. 7a, b). Compared with the trastuzumab treatment group, the roblitinib treatment group showed decreased tumor volume. The combination of trastuzumab and roblitinib revealed a synergistic antitumor effect in trastuzumab-resistant breast cancer (Fig. 7c–h). As indicated by IHC staining, FGFR4 inhibition significantly decreased the levels of p-GSK-3β, active β-catenin, SLC7A11, FPN1, and the cell proliferation marker Ki67 (Fig. 7i and Supplementary Fig. 8a–c).

To better validate the efficacy of roblitinib, patient-derived xenograft (PDX) and organoid (PDO) models were established (Fig. 7j). Among them, one successful culture model was derived from a non-pCR breast cancer tissue specimen obtained after trastuzumab-based neoadjuvant therapy, which reflected the biological characteristics of intrinsic anti-HER2 resistant breast cancer. Another model representing acquired anti-HER2 resistant breast cancer was established from a patient with liver metastasis discovered eighteen months after adjuvant trastuzumab treatment. The tumor volume and weight of tissues from the combination treatment (trastuzumab and roblitinib) group were significantly decreased compared with those of the vehicle, trastuzumab, and roblitinib groups (Fig. 7k–n). Moreover, we examined the toxicity of the drug to mouse livers (Supplementary Fig. 8d, e). Next, we evaluated the effect of roblitinib on overcoming anti-HER2 resistance in organoids (Fig. 7q, r). To rule out the possibility of contamination from liver cells and normal mammary epithelial cells, the patient-derived organoids were validated as HER2-overexpression breast cancer organoids before experiments (Supplementary Fig. 8f). The combination of trastuzumab and roblitinib remarkably attenuated growth in established trastuzumab-resistant breast cancer organoids (Fig. 7s). Together, these findings indicated that FGFR4 inhibition could restore sensitivity to anti-HER2 in resistant breast cancer cells. The combination of anti-HER2 and anti-FGFR4 might be a broadly effective therapy against both intrinsic and acquired resistant breast cancer.

## Discussion

Based on the occurrence time of treatment failure, anti-HER2 resistance can be classified into intrinsic (primary) and acquired (secondary) resistance[49]. For patients with early-stage HER2-positive breast cancer receiving adjuvant therapy, intrinsic resistance refers to recurrence or metastasis developing within 12 months of anti-HER2 treatment, while acquired resistance refers to recurrence or metastasis developing 12 months after the end of anti-HER2 adjuvant therapy. For advanced HER2-positive breast cancer, intrinsic resistance refers to progression within 3 months after first-line treatment of metastatic disease, while acquired resistance refers to disease progression after 3 months of anti-HER2 administration. Therefore, it is of great importance to find a target that is effective against both intrinsic and acquired anti-HER2 resistance in HER2-positive breast cancer. Recently, a functional CRISPR screening system has been acknowledged as a powerful discovery platform that enables researchers to efficiently identify potential therapeutic targets in multiple diseases[50–53]. In this study, we performed genome-wide CRISPR/Cas9-based loss-of-function screening under both in vitro and in vivo conditions to identify genes that drive anti-HER2 resistance in breast cancers. Deep analysis uncovered *FGFR4* as an essential gene for

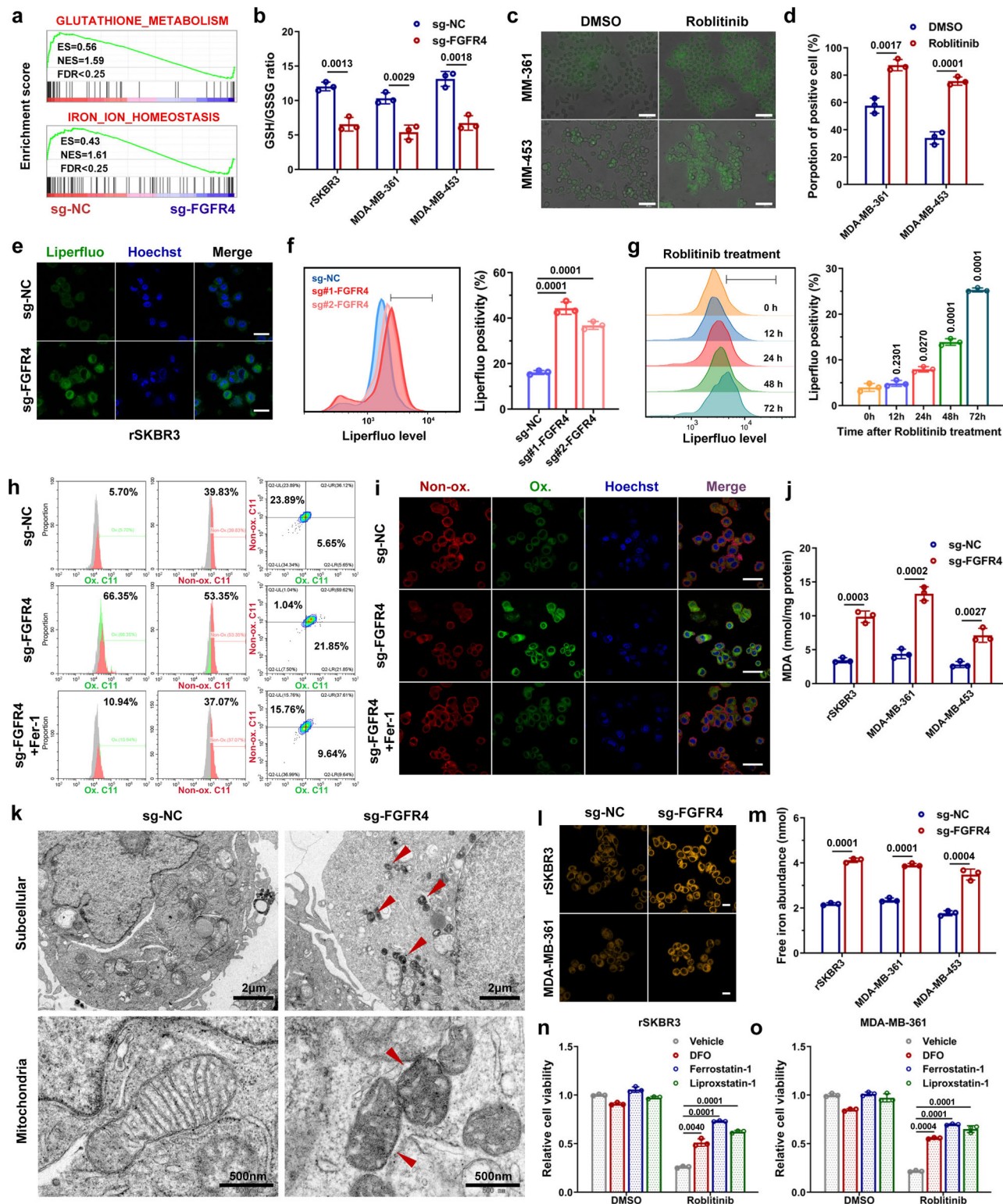

cellular survival following pharmacological HER2 signal blockade. As a member of the TKI family, FGFR4 is a druggable target with a first-in-class highly selective and potent inhibitor, roblitinib. Validated in both intrinsic and acquired trastuzumab-resistant HER2-positive cell lines and cell-derived xenograft models, inhibition of FGFR4 dramatically increased cell and tumor susceptibility to anti-HER2 therapy. Furthermore, PDX and PDO models were established to better examine the efficacy of roblitinib in anti-HER2 resistant breast cancer. The PDX model

retains most of the histopathology and molecular biology of the tumors which has good predictive value for treatment response[54]. The PDO model is derived from a group of cancer stem cells from patients and highly reproduces the clinical response by maintaining the tumor phenotype and genotype, which is efficient for preclinical drug discovery and validation[55,56]. A synergistic effect of roblitinib with trastuzumab was observed in PDX/PDO models originating from non-pCR HER2-positive breast cancer tissue after anti-HER2 based neoadjuvant therapy (intrinsic resistance)

**Fig. 5 Ferroptosis is triggered following FGFR4 inhibition in anti-HER2 resistant breast cancer cells. a** GSEA displaying correlations between FGFR4 expression and biological process in anti-HER2 resistant breast cancer cells. **b** The glutathione (GSH)-to-oxidized glutathione (GSSG) ratio was evaluated in anti-HER2 resistant breast cancer cells. **c**, **d** Liperfluo staining visualized lipid ROS in cells after treatment with roblitinib. Scale bar = 50 μm. **e–f** Laser confocal imaging and flow cytometry analysis of Liperfluo staining revealed an elevation in lipid ROS after FGFR4 inhibition. Scale bar = 20 μm. **g** Lipid ROS gradually accumulated on the cytomembrane after roblitinib treatment, as assessed by flow cytometry at specific time points. **h** The ratio of oxidized to nonoxidized lipids was assessed by flow cytometry following C11-BODIPY probe staining in rSKBR3 cells. **i** Confocal microscopy visualized the alterations in lipid peroxidation in rSKBR3 cells after C11-BODIPY probe staining. Scale bar = 30 μm. **j** The level of MDA (lipid peroxidation product) was significantly increased in FGFR4-deficient cells. **k** Transmission electron microscopy revealed a distinctive morphological feature of ferroptosis (shrunken mitochondria with increased membrane density) in cells subject to FGFR4 inhibition. **l**, **m** The level of intracellular ferrous ions ($Fe^{2+}$) was measured by FerroOrange probes and colorimetric iron assays. Scale bar = 10 μm **n**, **o** Cell death caused by roblitinib was partly rescued by cotreatment with the specific ferroptosis inhibitors ferrostatin-1 (1 μM), liproxstatin-1 (500 nM) or the iron chelator deferoxamine (DFO, 100 μM). Data in **b**, **d**, **f**, **g**, **j**, **m**, **n**, and **o** were presented as mean ± S.D., n = 3 biologically independent samples. Data were analyzed by two-sided Student's *t* test in **b**, **d**, **f**, **g**, **j**, **m**, and one-way ANOVA adjusted for multiple comparisons in **n**, **o**. Source data are provided as a Source Data file.

and liver metastasis of HER2-positive breast cancer tissue (acquired resistance). As the efficacy of anti-FGFR4 therapy is positively correlated with FGFR4 expression in tumor cells[35], we also examined the FGFR4 expression level in metastatic lesions of breast cancer obtained at our cancer center. Upon evaluation by IHC staining, metastatic tumors had higher FGFR4 expression than primary breast cancers, especially in brain metastases. This means that, as a small-molecule inhibitor that crosses the blood-brain barrier, roblitinib holds promise in treating patients with breast cancer brain metastases. These results provide us with a strategy to reverse anti-HER2 resistance via FGFR4 inhibition.

FGFR4 consists of an extracellular region composed of three immunoglobulin-like domains, a single hydrophobic membrane-spanning segment, and a cytoplasmic tyrosine kinase domain[57]. The extracellular domain of FGFR4 interacts with fibroblast growth factors, setting in motion a cascade of downstream signaling through phosphorylated kinase activity. For example, FGFR4 phosphorylates and activates STAT3 to promote tumor-extrinsic immune evasion in various cancers[58]. As a key node in the YAP/Hippo pathway, MST1/2 are phosphorylated in an FGFR4 kinase activity-dependent manner, resulting in the proliferation and metastasis of cancer cells[59]. Additionally, FGFR4 regulates cancer cell differentiation and metastasis in luminal subtype breast cancer which involves in hypoxia tolerance, glycolysis metabolism, and EMT process[60]. In our study, FGFR4 phosphorylated GSK-3β to activate β-catenin/TCF-4 signaling and drive anti-HER2 resistance in breast cancer. Ferroptosis, a non-apoptotic cell death caused by reactive oxygen species (ROS) and iron ion, was observed after anti-FGFR4 treatment. Based on previous studies, ferroptosis is greatly induced after anti-HER2 treatment in breast cancer[61,62]. Therefore, combined inhibition of FGFR4 and HER2 may trigger synergistic therapeutic effects of ferroptosis. In fact, several widely used antitumor agents have been found to inhibit cancer cells by inducing ferroptosis, which is a strategy for overcoming drug resistance[63,64]. For example, sorafenib (a Raf-1, B-Raf, and VEGFR multi-kinase inhibitor) treatment results in ferroptosis by blocking system Xc− in hepatocellular carcinoma[65]. Olaparib, a PARP inhibitor, promotes ferroptosis by impairing SLC7A11 in BRCA-mutant ovarian cancer[66]. The key mechanism inducing ferroptosis is excessive ROS production and labile iron pool accumulation in cells[47]. As expected, we found that FGFR4 inhibition could reduce the generation rate of glutathione and the transport efficiency of $Fe^{2+}$ efflux via transcriptional activation of *SLC7A11* and *FPN1* expression. As a key subunit of the amino acid transporter system Xc−, SLC7A11 takes up cystine in exchange for glutamate[67]. Cystine is crucial for glutathione synthesis, which is a major factor that combats the accumulation of ROS. FPN1 is a membrane protein involved in intracellular iron homeostasis and anti-ferroptosis effects by exporting ferrous ions from the

cytoplasm[68]. *FGFR4* is one of the few genes that can modulate both glutathione and iron ion metabolic pathways by upregulating these two pivotal genes.

Roblitinib is a reversible covalent small-molecule inhibitor of FGFR4 kinase activity. Unlike pan-FGFR inhibitors[69], roblitinib is highly selective for FGFR4 without off-target effects. A phase II clinical trial (NCT02325739) of roblitinib was completed and showed promising clinical efficacy and a favorable safety profile in hepatocellular carcinoma[35]. This inspiring result will greatly support the design of future clinical trials in the treatment of anti-HER2 resistant breast cancer. It should be noted that some single nucleotide polymorphism (SNPs) of *FGFR4* have great clinical significance. The *FGFR4* Arg388 allele exists in 51% of breast cancer patients and is related to worse progression-free survival[70]. The *FGFR4* Y367C activating mutant was discovered in MDA-MB-453 cells, a HER2-positive breast cancer cell line that is extremely sensitive to roblitinib. However, this mutation was not found by next-generation sequencing of 522 breast cancer samples[71]. Further studies should focus on the therapeutic effect of roblitinib in preclinical models with different *FGFR4*-SNPs before clinical trials. In addition, we only confirmed the impact of FGFR4 on tumor cells at the current stage. Due to the important role if immune system, future study should focus on the immune and stromal microenvironment in HER2-positive breast cancer.

In conclusion, our study illustrates that FGFR4 confers anti-HER2 resistance by attenuating ferroptosis in breast cancer (Supplementary Fig. 9). These results highlight a mechanism of anti-HER2 resistance and provide a strategy for overcoming resistance by FGFR4 inhibition in recalcitrant HER2-positive breast cancer.

## Methods

**Patient sample collection**. This study was approved by Institutional Research Ethics Committee of Sun Yat-sen University Cancer Center and conducted under the guidance of the Declaration of Helsinki. Samples of HER2-positive primary invasive carcinoma of the breast treated with anti-HER2 based therapy were collected from 332 patients between 2010 and 2020 in Sun Yat-sen University Cancer Center. After obtaining tissue samples during surgery, formalin fixation and paraffin embedding were performed using standard methods. Breast tumor tissue cores were collected from each patient and used to construct a tissue microarray for further validation. The molecular subtypes were determined by immunohistochemistry (IHC), and HER2 status was further confirmed by fluorescence in situ hybridization (FISH) if the IHC result was intermediate positive. Patients without detailed active follow-up were excluded. Overall survival (OS) was defined as the time from the date of diagnosis to the date of either death from any cause or last follow-up. Recurrence-free survival (RFS) was defined as the time from the date of diagnosis to the date of either first recurrence or last follow-up. For specimens from neoadjuvant therapy patients, samples of HER2-positive tumors were obtained from 36 breast cancer patients before they underwent preoperative neoadjuvant chemotherapy with a trastuzumab-based regimen and taxanes according to NCCN guidelines. Pathological complete response (pCR) was defined as the absence of invasive neoplastic cells upon microscopic examination of the primary tumor during surgery. Response Evaluation Criteria in Solid Tumor version 1.1 (RECIST

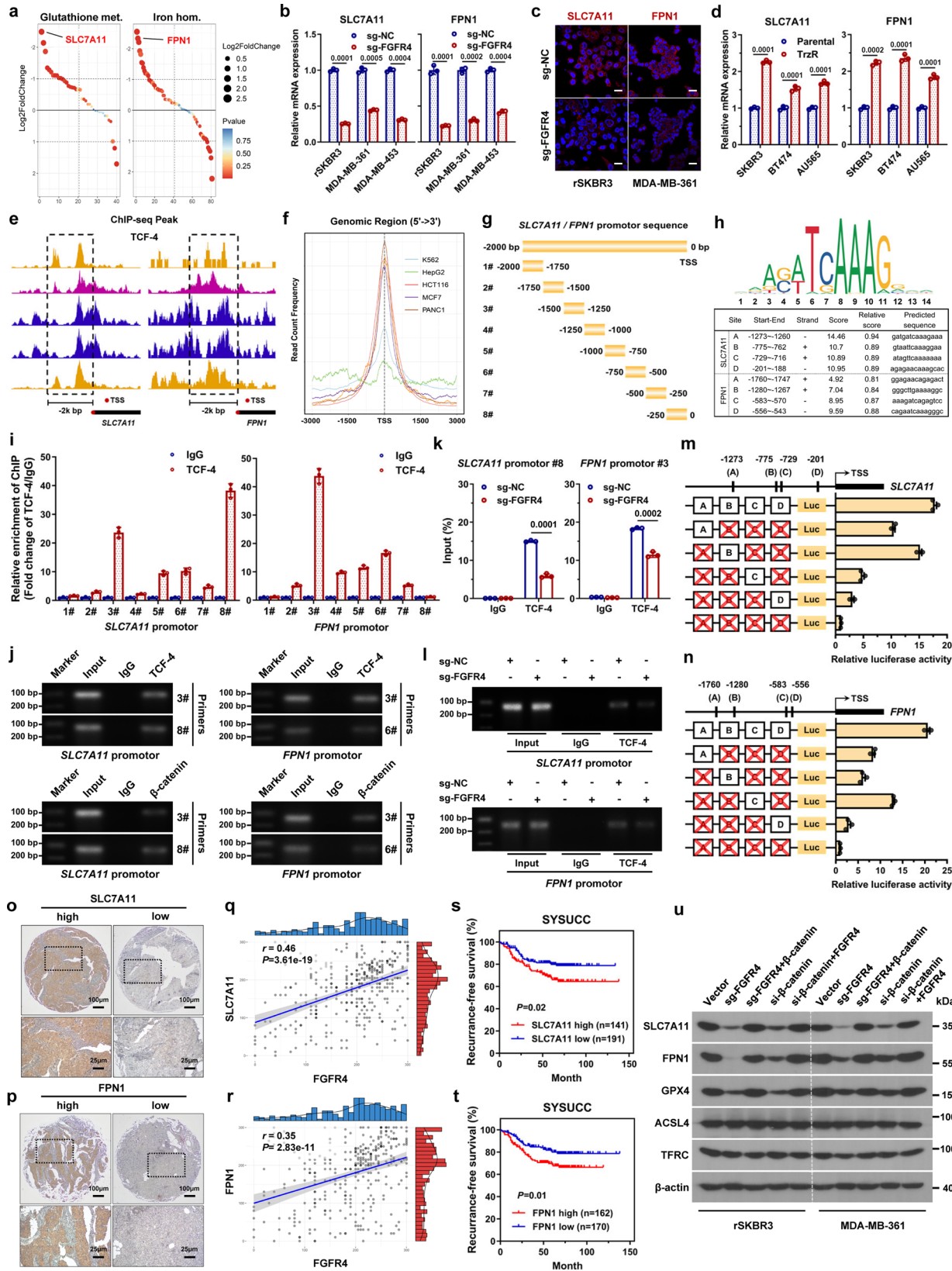

1.1) was used as the criteria to evaluate the tumor response to therapeutic regimens. Informed consent was obtained from all subjects.

**Cell lines and reagents.** Human breast cancer cell lines (MDA-MB-453, MDA-MB-361, SKBR3, BT474, AU565, MCF-7, T47D, ZR-75-1, MDA-MB-231, BT549, MDA-MB-468, and SUM-159), normal mammary epithelial MCF-10A cell line,

and HEK293T cell line were purchased from the American Type Culture Collection and cultured following the manufacturer's instructions. rSKBR3, rBT474, and rAU565 cells were developed by three-month exposure and selection with 1 μM trastuzumab. These acquired trastuzumab-resistant cell lines were continuously cultured in the regular medium with trastuzumab which was withdrawn four weeks before experiments. Cell lines were passaged for no more than six months and were authenticated by STR analysis. No mycoplasma infection was found for any cell

**Fig. 6 FGFR4 accelerates cystine uptake and Fe²⁺ efflux via the β-catenin/TCF4-SLC7A11/FPN1 axis. a** Gene rank based on fold-change in two pathways (glutathione metabolism and iron ion homeostasis). **b** *SLC7A11* and *FPN1* mRNA expression levels were validated by qPCR analysis. **c** SLC7A11 and FPN1 protein expression was evaluated by immunofluorescent staining visible under a laser confocal microscope. Scale bar = 20 μm. **d** *SLC7A11* and *FPN1* mRNA expression levels were upregulated in trastuzumab-resistant cells compared to their respective parental cells. **e** ChIP-seq analysis revealed TCF-4 binding peaks on the promoter regions of *SLC7A11* and *FPN1* according to the ENCODE database. **f** The read count frequency of genomic regions by ChIP-seq analysis. **g** Primers were designed according to the promoter sequences of the *SLC7A11* and *FPN1* loci. **h** TCF-4 binding motif and sequences as predicted by the JASPAR website. **i** Enrichment of *SLC7A11* and *FPN1* promoter fragments using an antibody against TCF-4 was assessed by ChIP-qPCR analysis. **j** Agarose electrophoresis analysis of PCR products of the *SLC7A11* and *FPN1* promoter after ChIP-PCR assays. **k, l** Enrichment of *SLC7A11* and *FPN1* promoter fragments was decreased after FGFR4 inhibition, as detected by ChIP-qPCR and agarose electrophoresis analysis. **m, n** Luciferase reporter assays for various deletions of exact binding elements in HEK-293T cells. **o, p** Representative IHC staining images showing high or low expression of SLC7A11/ FPN1 in 322 HER2-positive breast cancer tissues from SYSUCC. **q, r** Correlation between FGFR4 and SLC7A11/FPN1 expression in HER2-positive breast cancer tissues from SYSUCC. The two-sided Spearman test was used. **s, t** Kaplan-Meier analysis of the recurrence-free survival of HER2-positive breast cancer patients in the SYSUCC cohort with high or low SLC7A11/FPN1 expression. **u** Western blot assays showing that the levels of the indicated proteins were altered in trastuzumab-resistant HER2-positive breast cancer cells after FGFR4 inhibition; these levels were rescued by supplementation with β-catenin. Data in **b**, **d**, **i**, and **k** were presented as mean ± S.D., *n* = 3 biologically independent samples. Data were analyzed by hypergeometric test and adjusted for multiple comparisons in **a**, two-sided Student's *t* test in **b**, **d**, **k**, log-rank test in **s**, **t**. Source data are provided as a Source Data file.

line. Trastuzumab, pertuzumab, trastuzumab emtansine was obtained from Roche (Basel, Switzerland). The small-molecule FGFR4-selective inhibitor roblitinib (FGF-401, HY-101568) was purchased from MedChemExpress. Ferrostatin-1 (GC10380), liproxstatin-1 (GC15681), and DFO (GC30151) were purchased from GLPBIO. 5-Azacytidine (HY-10586) and trichostatin A (HY-15144) were purchased from MedChemExpress. Z-DEVD-FMK (T6005) and tucatinib (T2364) were purchased from TOPSCIENCE.

**Pooled CRISPR/Cas9 library in vitro and in vivo screening**. A total of 240 million trastuzumab-resistant rSKBR3 cells were seeded into twenty 15 cm plates and infected with the pooled GeCKOv2.0 human lentiviral library (over 300x coverage) at an MOI of 0.3 to ensure that most cells took up only one stably short guide RNA (sgRNA). Puromycin (1 μg/ml) was added to the cells at 24 h after infection and maintained in the cultures for 4 days. Afterward, cells were split into four groups at equal densities. Of these four groups, two were used for in vitro screening and two for in vivo screening. Mutant trastuzumab-resistant rSKBR3 cells were treated with vehicle or trastuzumab (1 μM for in vitro screening; 20 mg/ kg for in vivo screening) for 2 weeks. After drug selection, cells and tumors were harvested in each group, and genomic DNA was isolated using a DNA extraction kit (Omega, D3396). Sequences of sgRNAs were amplified by PCR, and the products were purified prior to sequencing. Resistant genes were identified from sgRNA screen sequencing results using MAGeCK analysis[72]. MAGeCK algorithm can prioritize the resistant genes by comparing the sgRNAs in the trastuzumab-treated cells/tumors to those in the vehicle-treated cells/tumors. Briefly, the read counts of each sgRNA from different samples were normalized to adjust for the effect of library sizes and read count distributions. Resistant genes are afterward identified by looking for genes whose sgRNAs are ranked consistently higher using robust rank aggregation (RRA). Genes with smaller RRA value ranked higher in the knockout screening.

**RNA isolation, quantitative real-time PCR (qPCR), and sequencing analysis**. TRIzol reagent (Invitrogen) was used to extract total RNA from cells and tissues. After reverse transcription was completed, the RNA expression level was examined by qPCR in triplicate on a Bio-Rad CFX96 using the SYBR Green kit (Takara, RR420A). Primer sequences are shown in Supplementary Table 5. Total RNA was extracted from sg-NC or sg-FGFR4 group of rSKBR3 breast cancer cells (2 × 10⁶) using TRIzol reagent (Invitrogen) for sequencing analysis. After the assessment of RNA integrity, the mRNA library was generated and sequenced by CapitalBio Technology (Beijing, China). The NEB Next Ultra RNA Library Prep Kit for Illumina (NEB) was used to construct the libraries for sequencing. All next-generation sequencing experiments were run on an Illumina NovaSeq sequencer (Illumina). Differentially expressed genes were subsequently screened out and identified with the criterion of |log2FC|> 1 and FDR < 0.05. DAVID tool was used to conduct Gene Ontology and Kyoto Encyclopedia of Genes and Genomes (KEGG) pathway enrichment analysis. Gene set enrichment analysis (GSEA) was applied to identify the significantly enriched pathways between the two groups using the GSEA 4.1.0 desktop application.

**Western blot analysis**. Total protein was extracted using RIPA lysis buffer with protease inhibitor cocktail. The protein concentrations were normalized with a BCA assay kit (Thermo Fisher, USA). Proteins from each group were loaded onto SDS-PAGE gels and separated before they were transferred to PVDF membranes (Millipore). The membrane was incubated with primary antibody against each target protein at 4 °C overnight. Afterward, the membrane was incubated with secondary antibody at room temperature for 1 h. Immunoblots were developed using a chemiluminescent reagent (Beyotime, China) according to the manufacturer's instructions[73]. The antibodies used are listed in Supplementary Table 6.

**Immunohistochemistry (IHC) and Hematoxylin-eosin (H&E) staining**. Paraffin-embedded tissues were used to conduct IHC staining. Section slides with tissue were deparaffinized in xylene and rehydrated via graded ethanol (dilution from 100%, 95%, 85% and 75%). Blockage of endogenous peroxidase activity and antigen retrieval was performed before incubation with primary antibody at 4 °C overnight. Staining was then conducted with diaminobenzidine (DAB) substrate (Dako) after incubation with secondary antibody (HRP-conjugated) for 20 min at room temperature. Hematoxylin was used to stain sections after DAB treatment. The antibodies used are listed in Supplementary Table 6. The staining intensity of each section was scored as 0 (no staining), 1+ (weak staining), 2+ (moderate staining), or 3+ (strong staining), and the percentage of positive cells was determined from five different random regions. H-score method (score range, 0–300) was used to quantify the expression. H-score from 0 to 200 was considered as low expression and H-score from 201 to 300 was assigned as high expression. For H&E staining, the sections were immersed in hematoxylin for 3 min, washed with water for 30 min, and dyed with eosin for 3 min. The slides were covered with coverslips after dehydration in ethanol at different concentrations. The stained slides were imaged under an optical microscope (NIKON ECLIPSE 80i).

**Cell viability and colony formation assays**. Cell viability was assessed using the Cell Counting Kit-8 kit (Dojindo, Japan). Briefly, 1 × 10³ cells were seeded into 96-well plates. CCK-8 solution (10 μL) mixed with medium was added to each well on a specific day. After incubation at 37 °C for 2 h, the absorbance at 450 nm was measured using a microtiter plate reader. CompuSyn software was used to evaluate the synergistic effect of FGFR4 and HER2 inhibition. Synthetic lethal effect is indicated if a combination index is lower than value 1. For colony formation assays, 1 × 10³ cells were seeded in 24-well plates. After 24 h, drugs at different concentrations were added to specific wells to test the sensitivity of the cells. Cell colonies were fixed with methanol and stained with 0.3% crystal violet before they were photographed.

**Mammosphere formation assay**. Mammosphere formation assays were performed as previously described[74]. A total of 5 × 10³ cells were resuspended and plated on ultralow attachment six-well plates in DMEM/F-12 medium (Gibco) with B27 supplement (Gibco), basic fibroblast growth factor (bFGF) (Invitrogen), EGF (Invitrogen) and insulin (Sigma). Cells were cultured for 7 days, and mammospheres with a diameter >100 μM were counted.

**Construction of vectors and transfection**. For CRISPRi repression of FGFR4, we constructed sgRNAs targeting the *FGFR4* promoter sequence and dCas9-KRAB fusion protein using a lentivirus-based plasmid[75] (Addgene, #71236). For programmable m6A modification of intracellular *FGFR4* mRNA with a Cas13-directed methyltransferase[76], we constructed sgRNAs targeting the upstream *FGFR4* mRNA m6A sites using the pC0043-PspCas13b crRNA backbone (Addgene, #103854). Fusion of dCas13, METTL3, and METTL14 proteins was used as an m6A methyltransferase tool (Addgene, #155367). The sequences of the sgRNAs are listed in Supplementary Table 7. The *SLC7A11/FPN1* promoter region (−1 to −2000 from the transcriptional start site) was amplified using hot-start DNA polymerase (Takara, RR006A). To construct a promoter-luciferase reporter, we cloned the promoter fragment into the pGL3-basic vector. Site-specific mutation of the *SLC7A11/FPN1* promoter region luciferase reporter construct was generated using a Site-Directed Mutagenesis Kit (YEASEN). The sequences of all the vectors were validated by Sanger sequencing. Cells were transfected using Lipofectamine 3000 (Invitrogen).

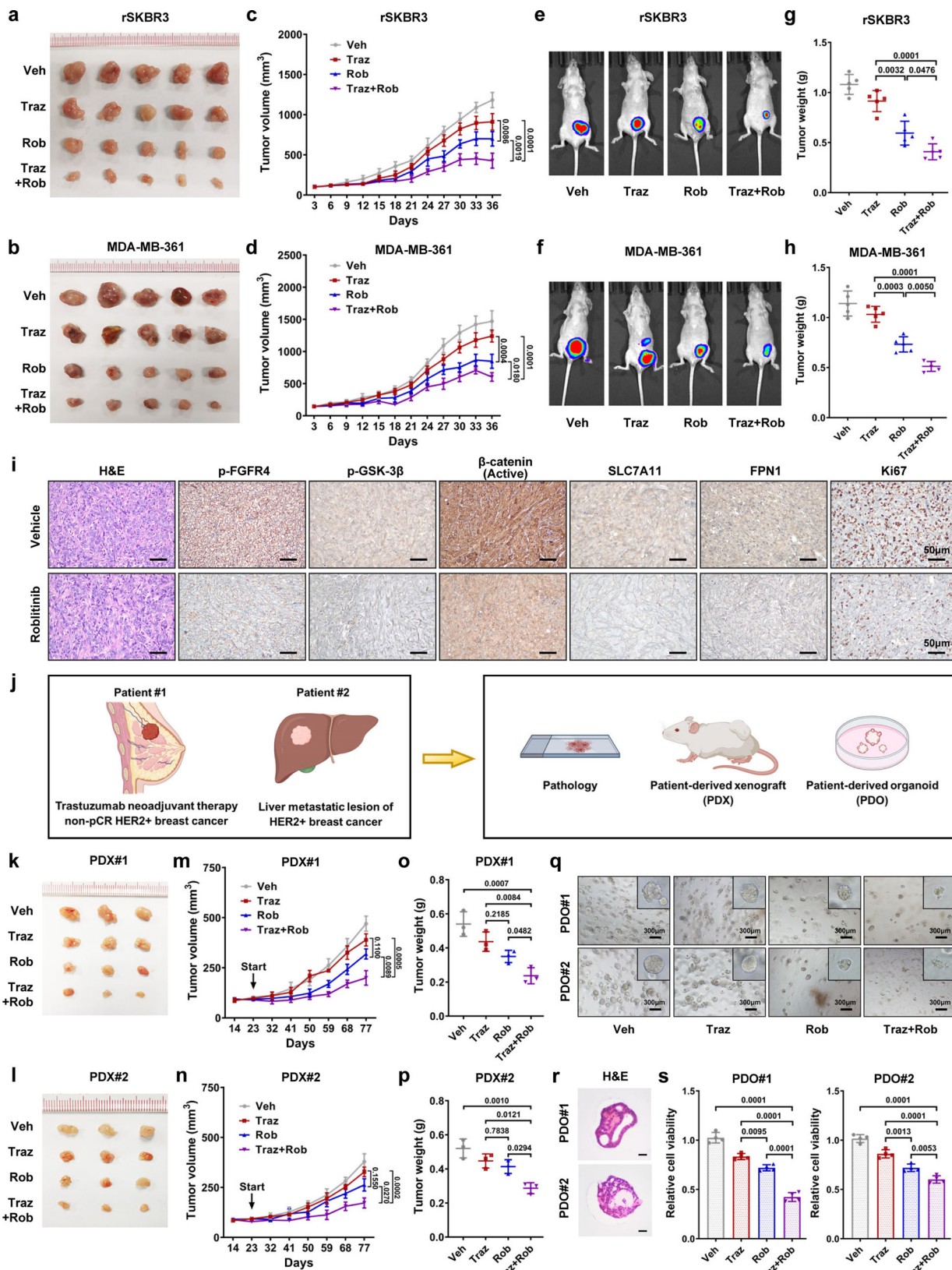

**RNA immunoprecipitation (RIP)**. Briefly, $2 \times 10^7$ cells were washed twice with PBS before they were harvested with RIP lysis buffer containing a protease inhibitor cocktail. The immunoprecipitation buffer was mixed with the lysed cells and incubated rotating at 4 °C overnight. The RNA-protein complexes were extracted with magnetic beads, and proteinase K was used to digest the protein. RNAs were purified before qPCR analysis. The RIP experiment was conducted using an RNA-Binding Protein Immunoprecipitation Kit (Millipore).

**Chromatin immunoprecipitation (ChIP)**. ChIP assays were performed as described previously using a ChIP assay kit (Bersin Bio) according to the instructions[77]. $2 \times 10^7$ cells were washed with PBS twice and collected after centrifugation. 1% formaldehyde was used for DNA crosslink at room temperature. The precipitation was washed and applied to lysis buffer to extract the nuclear contents. The DNAs were further fragmented and enriched with antibodies. The mixture was further inverse-crosslinked at 65 °C overnight and the enriched DNA

**Fig. 7 Patient-derived models revealed the efficacy of FGFR4 inhibitor in both intrinsic and acquired anti-HER2 resistant breast cancers.** Tumor-bearing mice established by rSKBR3 and MDA-MB-361 anti-HER2 resistant cells were treated with vehicle, trastuzumab (20 mg/kg, intraperitoneal administration), roblitinib (30 mg/kg, oral administration), or a combination of both agents. **a**, **b** Photographs of harvested primary tumors. **c**, **d** The volume of the tumor was recorded every 3 days, and the tumor growth curves were plotted. **e**, **f** Representative bioluminescence of mice taken at day 36. **g**, **h** The weights of the harvested xenograft tumors. Data were presented as mean ± S.D., $n = 5$ in each group. **i** Representative H&E and IHC staining images of MDA-MB-361 cell-based tumors from the vehicle and roblitinib groups. **j** Schematic diagram of the establishment of patient-derived xenografts (PDXs) and patient-derived organoids (PDOs) from intrinsic and acquired trastuzumab-resistant breast cancer tissues in SYSUCC. **k**, **p** In vivo experiments of PDX models treated with vehicle, trastuzumab (20 mg/kg, intraperitoneal administration), roblitinib (30 mg/kg, oral administration), or a combination of both agents. **k**, **l** Photographs of harvested PDX tumors. **m**, **n** The volume of the tumor was recorded every 9 days, and the tumor growth curves were plotted. **o**, **p** The weights of the harvested PDX tumors. Data were presented as mean ± S.D., $n = 3$ in each group. **q** Representative images of patient-derived organoids treated with vehicle, trastuzumab, roblitinib, or a combination of both agents. **r** Histological images of established patient-derived organoids. Scale bar = 50 μm. **s** Cell viability assays of patient-derived organoids treated with the corresponding agent. Data were presented as mean ± S.D., $n = 4$ biologically independent samples. Data were analyzed by one-way ANOVA adjusted for multiple comparisons in **c**, **d**, **g**, **h**, **m–p**, **s**. Source data are provided as a Source Data file.

---

was extracted. The amount of DNA pulled down was determined using qPCR analysis. The sequences of the primers used in the ChIP-PCR assays are listed in Supplementary Table 5.

**Luciferase reporter assay**. Cells were seeded into 24-well plates and transfected with luciferase reporter vectors as described above. After 48 h, the cells were lysed and tested using a dual-luciferase reporter assay kit (Promega, Madison, WI) according to the manufacturer's instructions. Independent experiments were conducted in triplicate.

**Methylated RNA immunoprecipitation (MeRIP) qPCR assay**. A MeRIP-qPCR assay was performed to detect m6A modifications of individual gene transcripts. Briefly, Protein A/G magnetic beads were prewashed and incubated with 5 μg of rabbit IgG or anti-m6A antibody (202003, Synaptic Systems) for 2 h at 4 °C with continuous rotation. Afterward, the antibody-bead complex was mixed with purified RNA and immunoprecipitation buffer containing RNase inhibitors (Beyotime, R0102). After immunoprecipitation, the proteins were digested by proteinase K, and RNA enrichment was examined by qPCR analysis.

**Dot blot assay**. First, isolated mRNA (200 ng for each group) was denatured by heating at 95 °C for 3 min, followed by chilling on ice immediately. Dilutions were spotted on an Amersham Hybond-N + membrane optimized for nucleic acid transfer (GE Healthcare). The membrane was crosslinked under UV light and washed with 1× PBST buffer. After the membrane was blocked with 5% nonfat milk in PBST, it was incubated with anti-m6A antibody (202003, Synaptic Systems) overnight at 4 °C. Then, the membrane was incubated with secondary antibody at room temperature for 1 h. Immunoblots were developed using a chemiluminescent reagent (Beyotime). The same 200 ng of mRNAs were spotted on the membrane, stained with 0.02% methylene blue in 0.3 M sodium acetate (pH 5.2) for 2 h and washed with RNase-free water for 1 h.

**RNA stability assay**. Breast cancer cells (rSKBR3 and MDA-MB-361) were plated in 12-well plates and exposed to actinomycin D (5 μg/mL, Sigma) for 0, 3, and 6 h. RNA was extracted at the indicated times and analyzed by qPCR. We calculated the mRNA half-life value by linear regression analysis.

**Glutathione (GSH) quantification and malondialdehyde (MDA) assay**. Total cellular glutathione levels were detected using the GSSG/GSH Quantification Kit (Dojindo, Japan) according to the instructions. For the MDA assay, the tumor cells were lysed and processed with a Lipid Peroxidation MDA assay kit (Beyotime). The absorbance at 532 nm was measured using a microtiter plate reader. All the values were normalized to the level of protein measured by the Micro BCA Protein Assay Kit (CWBIO).

**Lipid peroxidation assessed by C11-BODIPY and Liperfluo staining**. Tumor cells were seeded in six-well plates with glass bottoms before staining. For C11-BODIPY staining, 1 ml PBS containing 10 μM BODIPY 581/591 C11 (Thermo Fisher Scientific) was added to the cells. After incubation for 30 min at 37 °C, the cells were resuspended and analyzed immediately on a flow cytometer. To perform Liperfluo staining, cells were treated with Liperfluo (10 μM, Dojindo, Japan) for 1 h at 37 °C, digested by trypsin and immediately analyzed on a flow cytometer. The FITC channel was selected to measure fluorescence intensity.

**Iron quantification and FerroOrange staining**. The intracellular iron concentration was quantified using an iron assay kit (Abcam, ab83366) according to the manufacturer's instructions. FerroOrange (Dojindo, Japan) is a fluorescent probe that enables live-cell fluorescent imaging of intracellular $Fe^{2+}$. In this assay,

HBSS buffer solution containing 1 μmol/L FerroOrange reagent was added to the cells. After 1 h of incubation, intracellular iron ions were visualized under a confocal microscope.

**Annexin V/propidium iodide (PI), sytox staining, and LDH release assay**. Cell death process was detected by the Annexin V-FITC/PI Detection Kit (KeyGEN, China) followed by flow cytometry analysis. SYTOX Orange (Invitrogen, USA) staining was used to detect dead cells. Cell death was monitored by LDH release assay using the LDH Cytotoxicity Assay Kit (Beyotime, China) according to manufacturer's instructions.

**Patient-derived organoid (PDO) culture and drug sensitivity assay**. Fresh breast cancer tissues were resected and digested with 2 mg/ml collagenase (Sigma) on an orbital shaker at 37 °C for 2–6 h. After centrifugation and resuspension of the cells, they were seeded in Matrigel (Corning, 356255) into 24-well plates and covered by organoid medium consisting of advanced DMEM/F12 (Gibco), B27 supplement (Gibco), Hepes (Sigma), Glutamax (Gibco), nicotinamide (Sigma), Y-27632 (Abmole), N-acetylcysteine (Sigma), A83-01 (Tocris), SB202190 (Sigma), R-spondin 1 (R&D), basic fibroblast growth factor (bFGF) (Invitrogen), EGF (Invitrogen), Noggin (R&D), and penicillin/streptomycin (Gibco)[78]. After 1–3 weeks of culture, the numbers of breast cancer organoids were counted and passaged. For the drug sensitivity assay, CellTiter-Glo 3D Reagent (Promega, G9682) was used to measure ATP as a proxy for viable cells. After cells were lysed with 3D Reagent, the plates were shaken for 30 min at room temperature, and luminescence was detected.

**Animal experiments**. Four-week-old female BALB/c nude mice and NOD-SCID mice were purchased from Beijing Vital River Laboratories Animal Technology. Luciferase-labeled rSKBR3 and MDA-MB-361 cells ($1 \times 10^7$ cells) mixed with 1:1 Matrigel (Corning, 356237) were subcutaneously injected into the fat pads of mice. After a tumor was palpable, the mice were randomized into four groups (five mice per group), and they were treated with vehicle, trastuzumab (20 mg/kg, intraperitoneal administration), roblitinib (30 mg/kg, oral administration), or a combination of both drugs. The tumor volume was measured every 3 days, and the volume was estimated according to the formula volume = length × width$^2$/2. To visualize the tumor size, 150 mg/kg D-luciferin potassium salt (ATT Bioquest) was intraperitoneally injected into mice 10 min before imaging. The mice were sacrificed at the end of the experiment, and xenografts were resected, weighed, and photographed. To generate PDX models, fresh breast cancer samples from patients were subcutaneously inoculated into NOD-SCID mice. When the established PDXs reached ~500 mm$^3$, the tumors were transplanted to other mice. After a tumor was palpable, the mice were randomized to examine the therapeutic effects of trastuzumab and roblitinib. All mice were kept under specific-pathogen-free conditions in Animal Facility of Sun Yat-sen University Cancer Center. They were kept in an animal room with a 12-h light-dark cycle at a temperature of 20–22 °C with 40–70% humidity. The maximally permitted tumor diameter of 15 mm in any dimension was never exceeded. The tumor weight was not exceeded 10% of the mouse body weight. All animal procedures were approved by Institutional Animal Care and Use Committee of Sun Yat-sen University Cancer Center.

**Statistics and reproducibility**. All experiments were performed at least three times, immunofluorescence staining, immunohistochemical staining, transmission electron microscope, mammospheres formation assay, western blot assays, and DNA agarose gel blot representative images are shown. Data were analyzed using SPSS 25.0 software. Unpaired Student's $t$ test was used to analyze the differences between the two groups. Comparisons among multiple groups were analyzed using one-way ANOVA. Survival curves were described by Kaplan-Meier plots and compared with the log-rank test. The results are presented as means ± standard

deviation. All boxplots indicate median (center), 25th and 75th percentiles (bounds of box), and minimum and maximum (whiskers). $P < 0.05$ was considered statistically significant.

**Reporting summary**. Further information on research design is available in the Nature Research Reporting Summary linked to this article.

## Data availability

Expression and survival analyses for genes in breast cancer were performed using data obtained from the TCGA (https://portal.gdc.cancer.gov) and METABRIC (http://www.cbioportal.org). Data of CCLE (https://portals.broadinstitute.org/ccle) and CTRP (https://portals.broadinstitute.org/ctrp) were downloaded. The KM-plotter data is from the website (http://kmplot.com/analysis). The m6A modification landscape was obtained from m6A-Atlas (v.1.0) (http://180.208.58.66/m6A-Atlas/index.html). Pathway analysis was conducted with DAVID (version 6.8) (https://david.ncifcrf.gov/home.jsp) and GSEA software (version 4.1.0), whereas JASPAR software (version 2020) was used to predict the TCF-4 binding sequence and motif (http://jaspar.genereg.net). ChIP-seq datasets were downloaded and analyzed under guidance from the ENCODE database (https://www.encodeproject.org). The sequence data analyzed in this study are available in GEO database under the accession number GSE116335. All the other data are available within the article and its Supplementary Information. Source data are provided in this paper.

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

## Acknowledgements

This study was supported by the National Natural Science Foundation of China (82173366, Xiaoming Xie; 82073117, Hailin Tang). Some of the clipart icons appearing in Figs. 1a, 7j were created with BioRender (https://biorender.com).

## Author contributions

X.M.Xie, H.Tang, Z.S.Chen, and Y.Zou designed the study. Y.Zou, S.Zheng, X.H.Xie, Z.Tian, Y.Tang, and W.Tian collected the data. Y.Zou, X.Hu, Z.Tian, L.Yang, J.Xie, X.Deng, and Y.Zeng analyzed and interpreted the data. X.H.Xie, F.Ye, S.M.Yan, and Y.Kong provided clinical samples. Y.Zou and S.Zheng wrote the manuscript. X.Hu, Z.S.Chen, and Z.Tian edited the manuscript. X.M.Xie, H.Tang, and Z.S.Chen finally reviewed the manuscript. All authors read and approved the final manuscript.

## Competing interests

The authors declare no competing interests.
