## [Peer Review File · Nature Communications]

N6-methyladenosine regulated FGFR4 attenuates ferroptotic cell death in recalcitrant HER2-positive breast cancerEditorial Note: This manuscript has been previously reviewed at another journal that is not operating a transparent peer review scheme. This document only contains reviewer comments and rebuttal letters for versions considered at *Nature Communications*.

REVIEWER COMMENTS

Reviewer #1 (Remarks to the Author):

The authors have addressed my prior points and the manuscript is improved.

Reviewer #2 (Remarks to the Author):

A. Summary of the key results.

This study used an in vitro and in vivo genome-wide CRISPR screen to identify mediators of trastuzumab resistance in HER2+ breast cancer. They cultured HER2+ cell lines in vitro in the presence of high-dose (1 μ M) trastuzumab until resistance developed, and used one of these models for the screen. They found that knockout of FGFR4 sensitized to trastuzumab in vivo and in vitro, and upregulation of FGFR4 expression in trastuzumab-resistant cell lines. They then present a series of experiments to determine the mechanism underlying FGFR4 upregulation, as well as the mechanism by which FGFR4 promotes trastuzumab resistance. In the revision, the authors have included numerous new experiments addressing previous limitations, including showing that FGFR4 reduces sensitivity to dual-HER2 inhibition or T-DM1, and additional experiments to show that beta-catenin is involved in trastuzumab resistance.

B. Originality and significance.

While FGFR signaling has previously been implicated in trastuzumab resistance, the role of FGFR4 in modulating trastuzumab response via beta-catenin-TCF4-SLC7A11/FPN1 and ferroptosis attenuation is novel.

One major concern is that the dose of trastuzumab (1 μ M) used for the CRISPR screen and trastuzumab-resistant cell lines is exceedingly high. Most in vitro studies use 10-20 μ g/ml trastuzumab (\sim 0.14 μ M), which are considered to be "saturating" in vitro. In addition, the shifts in IC50 shown throughout are modest. Thus, the clinical/translational significance remains uncertain.

C. Data & methodology: validity of approach, quality of data, quality of presentation.

The data are much improved in the revised manuscript.

It should be noted that the NOD-SCID mice used for the PDX model lack NK cells and have impaired B and T cells, and thus the authors' claim in the rebuttal that "PDX model retains most of the immune microenvironmental characteristics of the tumors" is untrue.

D. Appropriate use of statistics and treatment of uncertainties.

Previous issues have been corrected.

E. Conclusions: robustness, validity, reliability.

Robustness, validity, and reliability have been improved in the revised manuscript.

F. Suggested improvements: experiments, further analyses, data for possible revision.

1. Additional methods details are still unclear:

a. Please describe the analysis of the CRISPR screen more clearly. Were the sgRNAs in the trastuzumab-treated cells/tumors compared to those in the vehicle-treated cells/tumors?

b. Supplementary figure 1c, d: Here, I assume the resistant cells were withdrawn from trastuzumab for 4 weeks. How long is the trastuzumab re-treatment in this figure, and what is the trastuzumab dose? The effects in this figure appear to be very modest. Have they been quantified?

c. Figure 1f: Have the patients in the METABRIC dataset been treated with trastuzumab?

d. Supplementary figure 2c: please clarify that FGFR4 expression was analyzed from tumors prior to treatment.

2. In addition, revisions to the text are needed.

Results/conclusions are often overstated or misstated. Examples include:

- a. Line 190: While the combination appears more effective than either drug alone, I would not call these results “synthetic lethal.”
- b. Line 202, 208, elsewhere: the shifts in IC50 throughout are quite modest. While FGFR4 may reduce sensitivity to anti-HER2 therapies, the effects do not seem sufficiently large to call it bona fide “resistance.”
- c. Line 301, 317: The data show that ERK/AKT phosphorylation are slightly decreased in resistant cell lines, but there are no data to indicate that these cells “lost dependence” on these pathways. It is quite possible that inhibition of ERK or AKT signaling would still be effective in these cells.
- d. Line 323: “dramatically restored sensitivity”—the results in supplementary figure 5e-f are not dramatic.

G. References: appropriate credit to previous work.

The authors appropriately give credit to previous work.

H. Clarity and context: lucidity of abstract/summary, appropriateness of abstract, introduction and conclusions.

Additions to the text require English language editing with regard to grammar and syntax.

Again, results are overstated in the abstract. Line 9: “FGFR4 inhibition *remarkably* enhances susceptibility”—please remove “remarkably”.

Reviewer #3 (Remarks to the Author):

The manuscript, now at Nature Communications, was improved through the addition of the following experiments:

- 1) Additions in Figure S6 and the very convincing time lapse videos have increased the quality of this manuscript.
- 2) Annexin V/PI, LDH release and sytox positivity in the presence and absence of a ferroptosis inhibitor were also added.
- 3) Specificity of immunohistochemistry were confirmed with convincing negative controls in HER2-positive breast cancer cells.

The overall data are very convincing and clearly of interest to the readership of Nature Communications.

In summary, the authors are congratulated to a significant contribution - I recommend publication of the manuscript.

Point-by-point Response to Reviewers' Comments

Reviewer #1 (Remarks to the Author):

The authors have addressed my prior points and the manuscript is improved.

Response: We thank the reviewer for the helpful comments that greatly improved our study.

Reviewer #2 (Remarks to the Author):

A. Summary of the key results.

This study used an in vitro and in vivo genome-wide CRISPR screen to identify mediators of trastuzumab resistance in HER2+ breast cancer. They cultured HER2+ cell lines in vitro in the presence of high-dose (1 μ M) trastuzumab until resistance developed, and used one of these models for the screen. They found that knockout of FGFR4 sensitized to trastuzumab in vivo and in vitro, and upregulation of FGFR4 expression in trastuzumab-resistant cell lines. They then present a series of experiments to determine the mechanism underlying FGFR4 upregulation, as well as the mechanism by which FGFR4 promotes trastuzumab resistance. In the revision, the authors have included numerous new experiments addressing previous limitations, including showing that FGFR4 reduces sensitivity to dual-HER2 inhibition or T-DM1, and additional experiments to show that beta-catenin is involved in trastuzumab resistance.

Response: We thank the reviewer for the valuable and constructive suggestions that greatly improved our study.

B. Originality and significance.

While FGFR signaling has previously been implicated in trastuzumab resistance, the role of FGFR4 in modulating trastuzumab response via beta-cenin-TCF4-SLC7A11/FPN1 and ferroptosis attenuation is novel.

One major concern is that the dose of trastuzumab (1 μ M) used for the CRISPR screen and trastuzumab-resistant cell lines is exceedingly high. Most in vitro studies use 10-20 μ g/ml trastuzumab (\sim 0.14 μ M), which are considered to be “saturating” in vitro. In addition, the shifts in IC50 shown throughout are modest. Thus, the clinical/translational significance remains uncertain.

Response: Thanks to the reviewer's comments. According to the previous published studies [1-6], full dose of trastuzumab (0.5~1 μ M) was used to ensure the complete

blockade of HER2 and maintain the concentration of antibody with short half-life. High concentration-induced resistant breast cancer cells maintained more stable drug resistance. Therefore, we used full dose of trastuzumab to generate resistant cell lines with strong and persistent drug resistance, as reported by references. The resistant cell lines still showed strong resistance at both low and high concentrations (0.1~1 μM) after withdrawn of trastuzumab for 4 weeks. To accurately evaluated the effect of trastuzumab at low dose (0.1 μM), we have added the previous result of CCK-8 assays in the revised manuscript (**Supplementary Figure 1d**). Thank you for this valuable feedback.

C. Data & methodology: validity of approach, quality of data, quality of presentation.

The data are much improved in the revised manuscript.

It should be noted that the NOD-SCID mice used for the PDX model lack NK cells and have impaired B and T cells, and thus the authors' claim in the rebuttal that "PDX model retains most of the immune microenvironmental characteristics of the tumors" is untrue.

Response: We highly appreciate this valuable suggestion from the reviewer. We greatly agreed that the statement claiming "PDX model retains most of the immune microenvironmental characteristics of the tumors" is not rigorous enough. Thus, such related terms have been removed in the revised manuscript.

D. Appropriate use of statistics and treatment of uncertainties.

Previous issues have been corrected.

Response: Thank you for your previous suggestions.

E. Conclusions: robustness, validity, reliability.

Robustness, validity, and reliability have been improved in the revised manuscript.

Response: Thank you for your previous suggestions.

F. Suggested improvements: experiments, further analyses, data for possible revision.

1. Additional methods details are still unclear:

a. Please describe the analysis of the CRISPR screen more clearly. Were the sgRNAs in the trastuzumab-treated cells/tumors compared to those in the vehicle-treated

cells/tumors?

Response: Thank you for this valuable advice. We have added the following description to the method section in the revised manuscript.

Resistant genes were identified from sgRNA screen sequencing results using MAGeCK (v0.5.4) software [7]. MAGeCK algorithm can prioritize the resistant genes by comparing the sgRNAs in the trastuzumab-treated cells/tumors to those in the vehicle-treated cells/tumors. Briefly, the read counts of each sgRNA from different samples were normalized to adjust for the effect of library sizes and read count distributions. Resistant genes are afterwards identified by looking for genes whose sgRNAs are ranked consistently higher using robust rank aggregation (RRA). Genes with smaller RRA value ranked higher in the knockout screening (Page 19, Line 608-615).

b. Supplementary figure 1c, d: Here, I assume the resistant cells were withdrawn from trastuzumab for 4 weeks. How long is the trastuzumab re-treatment in this figure, and what is the trastuzumab dose? The effects in this figure appear to be very modest. Have they been quantified?

Response: Thank you for the suggestion. The drug resistant cells were withdrawn from trastuzumab for 4 weeks before the experiments. Cells were treated with vehicle or trastuzumab (0.1 μ M) for 24 hours. Bright-field micrographs of the cells in the plate center showed the cell morphology instead of cell density (**Supplementary Figure 1c**). Unrepresentative micrographs and short treated time lead to modest effect observed in the figures. To accurately quantify the effect, we have added the previous result of CCK-8 assay in the current revised manuscript. The inhibitory effect of trastuzumab on parental and drug resistant cells was quantified after treated with trastuzumab (0.1 μ M) for 72 hours (**Supplementary Figure 1d**).

c. Figure 1f: Have the patients in the METABRIC dataset been treated with trastuzumab?

Response: We thank the reviewer for this comment. We rechecked the detailed information in the METABRIC dataset and found that not all the patients in the cohort were treated with anti-HER2 therapy. Unfortunately, the therapeutic information of individual patient was unavailable in the METABRIC dataset. Therefore, we only present the results of survival analysis of patients treated with anti-HER2 therapy in our center.

d. Supplementary figure 2c: please clarify that FGFR4 expression was analyzed from tumors prior to treatment.

Response: Thank the reviewer for this suggestion. The expression of FGFR4 was analyzed from tumors prior to treatment. We have added this explanation to the figure legend (**Supplementary Figure 2c**).

2. In addition, revisions to the text are needed.

Results/conclusions are often overstated or misstated. Examples include:

a. Line 190: While the combination appears more effective than either drug alone, I would not call these results “synthetic lethal.”

Response: Thank you for this comment. We conducted quantitative tests of synergy with the use of CompuSyn software [8]. Synergistic effect is indicated if a combination index (CI) is lower than value 1 (**Figure 2j-lower panel** , Page 6, Line 189). More detailed explanation had been added to make the statement clearer in the revised manuscript.

b. Line 202, 208, elsewhere: the shifts in IC50 throughout are quite modest. While FGFR4 may reduce sensitivity to anti-HER2 therapies, the effects do not seem sufficiently large to call it bona fide “resistance.”

Response: We thank the reviewer for this comment. We found that IC50 was reduced by 50%~90% after FGFR4 knockdown. According to the previous published studies, these changes can be described as reducing drug sensitivity. Future studies will be conducted to identify the best treating duration of anti-HER2 therapies. We changed the word to “reduced sensitivity” in the revision (Page 7, Line 209). Thank you for this kind remind.

c. Line 301, 317: The data show that ERK/AKT phosphorylation are slightly decreased in resistant cell lines, but there are no data to indicate that these cells “lost dependence” on these pathways. It is quite possible that inhibition of ERK or AKT signaling would still be effective in these cells.

Response: We agree with the reviewer's suggestion. We have changed the term “lost dependence” to the term “reduced dependence” in our revised manuscript (Page 10, Line 302).

d. Line 323: “dramatically restored sensitivity”—the results in supplementary figure 5e-f are not dramatic.

Response: Thank you for this advice. We have deleted this word in our revised manuscript (Page 10, Line 324).

G. References: appropriate credit to previous work.

The authors appropriately give credit to previous work.

Response: Thank you for your previous suggestions.

H. Clarity and context: lucidity of abstract/summary, appropriateness of abstract, introduction and conclusions.

Additions to the text require English language editing with regard to grammar and syntax.

Response: We have carefully proofread the manuscript to ensure the correct grammar and syntax before resubmission. Thank you for this kind remind.

Again, results are overstated in the abstract. Line 9: “FGFR4 inhibition *remarkably* enhances susceptibility”—please remove “remarkably”.

Response: Thank you for this advice. We have removed this word in our revised manuscript (Page 1, Line 9).

Thank you for the careful review and kind suggestions. We hope that these responses are adequate and would like to express our sincere gratitude to you for your time and considerations in reviewing our manuscript.

Reviewer #3 (Remarks to the Author):

The manuscript, now at Nature Communications, was improved through the addition of the following experiments:

- 1) Additions in Figure S6 and the very convincing time lapse videos have increased the quality of this manuscript.
- 2) Annexin V/PI, LDH release and sytox positivity in the presence and absence of a ferroptosis inhibitor were also added.
- 3) Specificity of immunohistochemistry were confirmed with convincing negative controls in HER2-positive breast cancer cells.

The overall data are very convincing and clearly of interest to the readership of Nature Communications.

In summary, the authors are congratulated to a significant contribution - I recommend publication of the manuscript.

Response: We thank the reviewer for the helpful comments that greatly improved our study.

References

1. Konecny GE, Pegram MD, Venkatesan N, Finn R, Yang G, Rahmeh M, et al. Activity of the dual kinase inhibitor lapatinib (GW572016) against HER-2-overexpressing and trastuzumab-treated breast cancer cells. *Cancer Res.* 2006;66(3):1630-9.
2. Paroni G, Bolis M, Zanetti A, Ubezio P, Helin K, Staller P, et al. HER2-positive breast-cancer cell lines are sensitive to KDM5 inhibition: definition of a gene-expression model for the selection of sensitive cases. *Oncogene.* 2019;38(15):2675-89.
3. Scaltriti M, Eichhorn PJ, Cortés J, Prudkin L, Aura C, Jiménez J, et al. Cyclin E amplification/overexpression is a mechanism of trastuzumab resistance in HER2+ breast cancer patients. *Proc Natl Acad Sci U S A.* 2011;108(9):3761-6.
4. Palomeras S, Diaz-Lagares Á, Viñas G, Setien F, Ferreira HJ, Oliveras G, et al. Epigenetic silencing of TGFBI confers resistance to trastuzumab in human breast cancer. *Breast Cancer Res.* 2019;21(1):79.
5. Choi HJ, Jin S, Cho H, Won HY, An HW, Jeong GY, et al. CDK12 drives breast tumor initiation and trastuzumab resistance via WNT and IRS1-ErbB-PI3K signaling. *EMBO Rep.* 2019;20(10):e48058.
6. Li X, Xu Y, Ding Y, Li C, Zhao H, Wang J, et al. Posttranscriptional upregulation of HER3 by HER2 mRNA induces trastuzumab resistance in breast cancer. *Mol Cancer.* 2018;17(1):113.
7. Li W, Xu H, Xiao T, Cong L, Love MI, Zhang F, et al. MAGeCK enables robust identification of essential genes from genome-scale CRISPR/Cas9 knockout screens. *Genome Biol.* 2014;15(12):554.
8. Chou TC. Drug combination studies and their synergy quantification using the Chou-Talalay method. *Cancer Res.* 2010;70(2):440-6.

REVIEWERS' COMMENTS

Reviewer #2 (Remarks to the Author):

The authors have sufficiently addressed all my concerns.

Point-by-point Response to Reviewers' Comments

Reviewer #2 (Remarks to the Author):

The authors have sufficiently addressed all my concerns.

Response: We sincerely thank the reviewer for the helpful comments that greatly improved our study.